# Confining single Er³⁺ ions in sub-3 nm NaYF₄ nanoparticles to induce slow relaxation of the magnetisation

Diogo A. Gálico [1], Emille M. Rodrigues[1], Ilias Halimi[1], Juho Toivola[2], He Zhao [3], Jiahui Xu [3], Jani O. Moilanen [2] ✉, Xiaogang Liu[3], Eva Hemmer [1,4] ✉ & Muralee Murugesu [1,4] ✉

Molecular systems known as single-molecule magnets (SMMs) exhibit magnet-like behaviour of slow relaxation of the magnetisation and magnetic hysteresis and have potential application in high-density memory storage or quantum computing. Often, their intrinsic magnetic properties are plagued by low-energy molecular vibrations that lead to phonon-induced relaxation processes, however, there is no straightforward synthetic approach for molecular systems that would lead to a small amount of low-energy vibrations and low phonon density of states at the spin-resonance energies. In this work, we apply knowledge accumulated over the last decade in molecular magnetism to nanoparticles, incorporating Er³⁺ ions in an ultrasmall sub-3 nm diamagnetic NaYF₄ nanoparticle (NP) and probing the slow relaxation dynamics intrinsic to the Er³⁺ ion. Furthermore, by increasing the doping concentration, we also investigate the role of intraparticle interactions within the NP. The knowledge gained from this study is anticipated to enable better design of magnetically high-performance molecular and bulk magnets for a wide variety of applications, such as molecular electronics.

Magnetic materials are ubiquitous in our daily lives, ranging from electric motors, cars and turbines to electronic devices, and their need is projected to increase exponentially in the future as we approach decarbonization[1]. Therefore, it is critical to develop lighter, stronger, and higher-performing magnetic materials. The magnet-like behavior of traditional bulk magnets often originates from the intimate interconnection and collective behavior of each individual spin center. To understand the overall macroscopic behavior of a magnet (e.g., slow magnetic relaxation dynamics), it is important to examine the basic units of bulk material at the atomic level, and shed light on the intrinsic properties of individual components (e.g., spin state of metal ions). Improved understanding of the unique role of these components at the atomic level makes it possible to tune and optimize their bulk

physicochemical properties, as well as to develop high-performing materials. For instance, in permanent magnets such as Nd₂Fe₁₄B and Sm₂Co₁₇ alloys, the incorporation of rare-earth (RE) ions provides significant magnetic anisotropy, leading to large magnetic coercivity and thus, magnetic hardness[2–6]. However, elucidating the exact contribution of each component at the atomic level remains a challenge.

As molecular model systems can provide insight into the spin and magnetic anisotropy, RE-based molecular systems, namely single-molecule magnets (SMMs), have been intensively developed over the past two decades[7–9]. Some of these systems exhibit magnet-like behavior of slow relaxation of the magnetization below their blocking temperature ($T_B$; below which, the material acts like a magnet), and giant coercivity that at times surpasses the coercive fields of

[1]Department of Chemistry and Biomolecular Sciences, University of Ottawa, Ottawa, ON K1N 6N5, Canada. [2]Department of Chemistry, Nanoscience Centre, University of Jyväskylä, P.O. Box 35, FI-40014 Jyväskylä, Finland. [3]Department of Chemistry, National University of Singapore, 3 Science Drive 3, Singapore 117543, Singapore. [4]Centre for Advanced Materials Research (CAMaR), University of Ottawa, Ottawa, ON K1N 6N5, Canada. ✉e-mail: jani.o.moilanen@jyu.fi; ehemmer@uottawa.ca; M.Murugesu@uottawa.ca

permanent magnets[7–9]. Mononuclear dysprosocenium complexes that act as SMMs above liquid nitrogen's temperature demonstrated remarkable magnetic performance can arise from a single RE center[10,11]. Although molecular systems can provide insight into the spin and magnetic anisotropy, the bulky ligands used to achieve a low coordination environment induce a high-density of molecular vibrations over a broad spectral range (from hundreds to a few thousand wavenumbers)[12,13]. The high-density of (low-energy) molecular vibrations increases the probability of the phonons coupling with (low-energy) electronic states, resulting in phonon-assisted relaxation which reduces the energy barrier ($U_{eff}$) and the overall magnetic performance of a system. Currently, the upper limit for relaxation barriers with existing compounds have already been reached[14]. Therefore, to further minimize phonon-assisted relaxation and improve the overall magnetic performance of a molecular magnet, it is critical to incorporate magnetic ions in a host matrix that not only provide a low vibrational density of states at the spin resonance energies but also rigidity at the molecular level that leads to a small amount of low-energy vibrations that can couple with low-energy (acoustic) phonons[15].

In contrast, nanoparticles (NPs), based on inorganic elements, provide an alternative ionic framework for the RE ions in which vibrational modes are in more discrete frequency ranges than in molecular compounds (see ESI for details)[16–27]. They also exhibit distinct vibrational modes, whose positions and intensities can be modulated by the nature of doped RE ions.

Additionally, their set crystalline lattice offers well-defined local high symmetry around metal centers, thus enabling symmetry consideration, a unique aspect in this study. Thus, NPs can be seen as interesting alternative host materials to investigate single-ion magnet (SIM) behavior at the atomic level with and without neighboring magnetic centers. The gained knowledge will be leveraged to gain a better understanding of classical bulk magnets and consequently lay the basis for the development of better performing bulk magnets. Small nanoparticles contain several hundreds of atoms which renders atomic-level analysis difficult. Thus, in an attempt to isolate one paramagnetic ion in a diamagnetic NP host, ultrasmall sub-3 nm NPs are targeted. Owing to their high chemical reactivity and behavior similarities, RE ions can be chemically "interchangeable" at the atomic level with their congeners within the crystalline lattice. This has been the method of choice for enhancing and harnessing luminescent properties in mixed-metal RE-based NPs (RE-NPs)[28,29]. In addition, low dopant concentrations allow for efficient dilution of $RE^{3+}$ ions at the atomic level. Recent achievements in the chemical synthesis of small and ultrasmall RE-NPs offer an excellent level of RE-ion/host tunability, being leveraged, for instance, into upconverting and near-infrared (NIR) emitting RE-NPs[30–32]. RE-NPs, such as alkali metal RE fluorides ($MREF_4$, e.g., M = Na, Li, K; RE = Y, Gd) doped with upconverting or near-infrared emitting $RE^{3+}$ dopant ions (e.g., Er/Yb, Tm/Yb, Nd, Ho and their combinations) have made strides in the field of novel upconverting and NIR-emitting nanoparticles[28–32].

To shed light on the magnetic contribution of individual components to bulk materials, we have focused on isolating a paramagnetic $RE^{3+}$ ion (namely $Er^{3+}$) through doping within a diamagnetic $NaYF_4$ NP host. Our ultimate goal is to sequester one $Er^{3+}$ ion *per* NP and probe its magnetic properties at the atomic level within this alternative framework for molecular compounds. Subsequently, by increasing the concentration of $Er^{3+}$ ions within the RE-NPs, we intend to systematically investigate the role of Er-Er interactions at various dopant concentrations; i.e., the influence of close $Er^{3+}$ ion proximity on the overall magnetic properties. As such, a slow relaxation mechanism can be probed using SQUID magnetometry at first at the individual atomic level and subsequently with the presence of neighboring paramagnetic ions by carefully controlling the percentage of dopant ions. Herein, we present the elucidation of the slow relaxation of the magnetization of

$Er^{3+}$ at the atomic level achieved through its meticulous dilution in a $NaYF_4$ NP. Observed magnetic relaxation dynamics at the single-ion level within the $NaYF_4$ NP represents the first report of SIM behavior in a RE-based nanoparticle. The knowledge gained from this unique study could unravel the untapped potentials of RE-based nanoparticles for applications ranging from memory storage at a single-atom level to fine-tuning high-performing hard magnets in a controlled fashion, which can be used for molecular electronic applications.

## Results and discussions

Ultrasmall (*ca* 2.7 nm, Fig. 1a) diamagnetic fluoride-type cubic-phase *α*-$NaYF_4$ NPs were chosen as host for $Er^{3+}$ ions acting as the paramagnetic component by replacing the diamagnetic $Y^{3+}$ ions in the crystalline lattice. Various $Er^{3+}$ nominal concentrations (1, 2, 4, 6, 8, and 10 mol%—hereafter, respective samples are named NPX with X being the $Er^{3+}$ nominal concentration) were probed to understand the role of Er-Er distances on the overall magnetic properties. Upon dilution of $Er^{3+}$ ions in these NPs at 1% molar concentration, we anticipate only one $Er^{3+}$ ion to be present per NP (Fig. 1b); thus, we target to probe the magnetic relaxation of the single $Er^{3+}$ ion at the atomic level in the rigid NP host.

RE-doped $NaYF_4$ NPs can be isolated in cubic *α*- or hexagonal *β*-phase. The Y centers in the *α*-phase adopt an eight-coordinate environment, while in the *β*-phase, two different nine-coordinate $C_{3v}$ sites are observed. More specifically, the *α*-phase of $NaYF_4$ can be described as $CaF_2$-type, with only one possible cation site of high symmetry ($O_h$), half occupied by $Na^+$ ions and half occupied by $RE^{3+}$ ions (Fig. 1c)[28]. In theory, the single high symmetry cubic coordination environment provided by *α*-$NaYF_4$ should eliminate the multiple relaxation processes originating from the different site symmetries of occupied sites that could be observed for *β*-phase due to the two different nine-coordinate $C_{3v}$ sites. In other words, *α*-phase is more straightforward to investigate than *β*-phase as there is only one possible high-symmetry site for cation. As previously mentioned, the main goal of this study is to prove that the knowledge obtained over the last decades with molecular compounds can be applied to nanoparticles; therefore, choosing a simpler system with an expected single-relaxation process is an exciting approach when translating the knowledge for the nanoparticle field for the first time. Although the pure $O_h$ symmetry stabilizes the isotropic ground state that is prone to the fast relaxation of the magnetization contrast to the anisotropic ground state[33], any distortions on the $O_h$ site can give rise to the magnetic anisotropy of $Er^{3+}$ ions[34]. With that said, some slight but non-negligible distortion around the $RE^{3+}$ coordination environment is expected given the slightly different ionic radii of the dopant and host $RE^{3+}$ ion (1.15 Å for $Y^{3+}$ and 1.14 Å for $Er^{3+}$)[35,36]. Moreover, decreasing the particle size to the nanoscale leads to higher surface tensions, which can result in a stabilization of the higher symmetry for these NPs—in some cases, even large enough to lead to a phase transition from an anisotropic to a more isotropic crystalline phase[37]. The $Er^{3+}$-doped *α*-$NaYF_4$ NPs ($NaYF_4$:$Er^{3+}$) were synthesized using the microwave-assisted thermal decomposition of $RE^{3+}$ oleate precursors, as described in previous work[38], resulting in ultrasmall NPs capped with oleate (−OA) groups. Regardless of the $Er^{3+}$ dopant concentration (1, 2, 4, 6, 8, or 10 mol%), all the synthesized NPs crystallized in the pure cubic (*α*)-phase of $NaYF_4$ (Fig. S1), exhibiting a spherical morphology of comparable sizes ranging from 2.5 to 2.7 nm (Figs. 1a, b and S2). Inductively coupled plasma (ICP) analysis (Table S1) was performed to confirm the $Er^{3+}$ ion concentration in the synthesized NPs.

The number of $Er^{3+}$ ions in each of the synthesized NPs was calculated as previously reported by Wilhelm and co-workers (see ESI)[39], indicating the presence of 1 (NP1), 2 (NP2), 4 (NP4), 7 (NP6), 9 (NP8), and 12 (NP10) $Er^{3+}$ ions *per* NP (Table S2). To further understand the distribution of $Er^{3+}$ ions within the NP, Monte Carlo simulations were performed (Figs. 1d, e and S4 and Table S3). For each NP composition, these simulations provided a mean $Er^{3+}$-$Er^{3+}$ distance within the

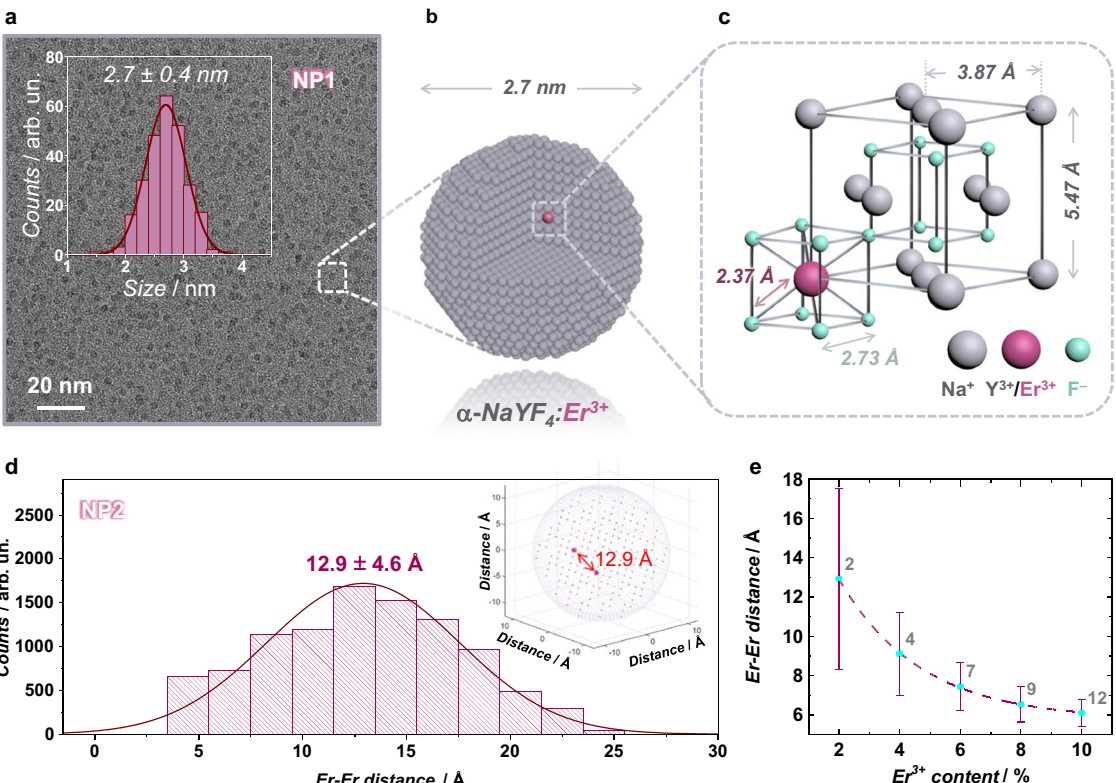

**Fig. 1 | Structural analysis for the NPs. a** Transmission electron microscopy (TEM) image and size distribution of $\alpha$-NaYF$_4$ NPs doped with 1 mol% Er$^{3+}$ (NP1). **b** Schematic representation of NP1 containing one Er$^{3+}$ ion. **c** Cubic lattice unit cell of NaYF$_4$ indicating the positions of Na$^+$, Y$^{3+}$, and F$^-$ host ions as well as Er$^{3+}$ dopants. The cubic lattice was adapted from ref. 14 for $\alpha$-NaYF$_4$. **d** Er$^{3+}$-Er$^{3+}$ distance distribution for NP2 calculated with Monte Carlo model (inset: Monte Carlo model of the number of Er$^{3+}$ ions and the mean Er$^{3+}$-Er$^{3+}$ distance for NP2). **e** Mean Er$^{3+}$-Er$^{3+}$ distance for the NPs as a function of the Er$^{3+}$ content. Gray labels indicate the number of Er$^{3+}$ ions within the NP as determined by ICP analysis. TEM images and Er$^{3+}$-Er$^{3+}$ distance distributions for all other compositions are given at the ESI.

diamagnetic host. For instance, for NP2 containing two Er$^{3+}$ ions *per* NP, the mean distance between the two Er$^{3+}$ ions was determined as 12.9 Å. Increasing the number of Er$^{3+}$ ions *per* NP resulted in the decrease of the mean distance, hence, approximation of the paramagnetic Er$^{3+}$ ions and yielding the shortest distance of 6.1 Å for NP10 (12 Er$^{3+}$ ions per NP).

To probe the intrinsic magnetic properties of the Er$^{3+}$ ions isolated within the $\alpha$-NaYF$_4$ host and how the intraparticle dipolar interactions could affect the slow relaxation of the magnetization, we have carried out in-depth direct current (dc) and alternating current (ac) magnetic susceptibility measurements on NPs with different Er$^{3+}$ doping concentrations. It is important to notice that thermogravimetric (TGA) analysis of the NPs (Fig. S3) confirmed a similar amount of residual hexane and oleate ligands at the surface of the NPs, allowing for normalization in relation to the mass for the magnetic data. Dc measurements are shown in Fig. 2a as mass susceptibility ($\chi_g T$) *vs.* temperature ($T$) from 300 K down to 1.8 K. For all the doping concentrations, an almost constant $\chi_g T$ value was observed from room temperature down to ~100 K. The decrease of the $\chi_g T$ values below this temperature is likely due to depopulation of the ligand field sublevels as often encountered in RE systems[40,41]. More importantly, for NP1, NP2, and NP4, $\chi_g T$ values at 1.8 K (Fig. 2c) increase linearly according to the anticipated number of Er$^{3+}$ ions *per* particle. This trend is indicative of negligible magnetic interactions between paramagnetic metal centers within the particles as the intraparticle Er$^{3+}$-Er$^{3+}$ distance is predicted to be above 9 Å for NP 2 and NP4, based on Monte Carlo simulations (Table S3). Whereas, increasing the Er$^{3+}$ concentration (above 6 mol%) resulted in a deviation from this linearity, thus, suggesting an increase in intraparticle dipolar magnetic interactions due to the greater amount of Er$^{3+}$ ions within the NPs and hence shorter Er$^{3+}$-Er$^{3+}$ distances. To further understand our observation, $\chi_g T$ *vs.* $T$ was also

collected with a field of 1 T (Fig. 2b, d). As can be noted, with a higher field, linearity at 1.8 K occurs up to NP8. Correlating the dc magnetic data with the mean Er$^{3+}$-Er$^{3+}$ distances within each NP (Fig. 1e) suggests that a distance close to and below 8 Å is the critical distance to increase the intraparticle dipolar interactions. The finding is supported by the calculated dipolar coupling between two Er$^{3+}$ ions residing in the crystal field (C.F.) generated by [Na$_8$Y$_4$ErF$_{24}$]$^-$ fragment (see ESI for details). The values of the calculated dipolar coupling parameters decrease with the increasing Er$^{3+}$–Er$^{3+}$ distance (r): $J_{dip} = -1.07$ cm$^{-1}$ ($r = 3.868$ Å), $J_{dip} = -0.38$ ($r = 5.470$ Å), $J_{dip} = -0.09$ ($r = 8.649$ Å), and $J_{dip} = -0.05$ ($r = 10.940$ Å). It is evident from the calculated data that the strength of the dipolar coupling at ~8 Å is one order of magnitude smaller than at ~3 Å. Because the dipolar interaction is inverse proportional to the third power of the distance between the magnetic centers, its magnitude and influence on the magnetic properties of magnetic sites decreases with increasing distance as proven by the data obtained for the studied NPs and previously reported SMMs[42–45]. Furthermore, this intraparticle dipolar interaction is anticipated to affect on the quantum tunneling of the magnetization (QTM) and relaxation times (ac data, *vide infra*).

To probe the magnet-like behavior of slow relaxation of the magnetization originating from a single Er$^{3+}$ ion level, ac susceptibility studies in the 0.1–1500 Hz range were performed with an oscillating field of $H_{ac} = 3.78$ Oe. In the absence of an applied static dc magnetic field, ac susceptibility signals were not observed. Such behavior is common for RE ions as they are prone to undergo QTM[46,47]. To minimize QTM and visualize slow relaxation of the magnetization, a static dc field can be applied. As such, to elucidate the optimal applied dc field, field-dependent relaxation dynamics were investigated at 1.9 K for each Er$^{3+}$ concentration (Fig. 3 left).

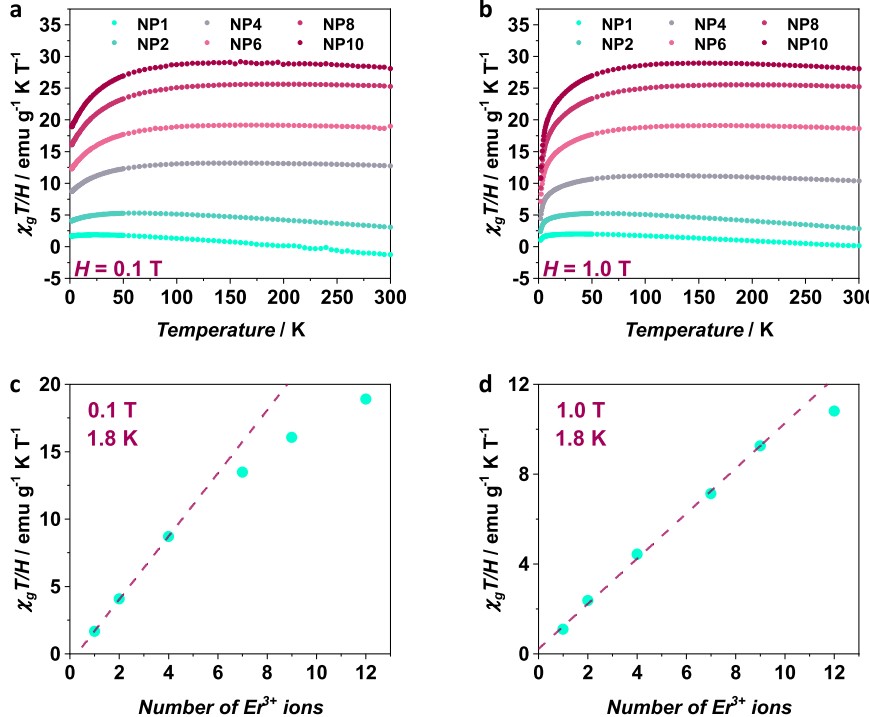

**Fig. 2 | Direct current (dc) magnetic susceptibility measurements for the NPs. a** Mass susceptibility ($\chi_gT/H$) as a function of temperature with an applied field of 0.1 T and (**b**) 1 T. **c** The number of $Er^{3+}$ ions dependence of the $\chi_gT/H$ value at 1.8 K with an applied field of 0.1 T and (**d**) 1 T. Source data are provided as a Source Data file.

Frequency-dependent signals in the out-of-phase $\chi''$ susceptibility were observed between 100–5000 Oe (and 400–5000 Oe for NP10), yielding a bimodal profile and indicating the presence of slow magnetic relaxation. Of these two frequency-dependent peak maxima, one is dominant at the high-frequency region (≥100 Hz; denoted as process 1) and the other at the low-frequency region (≤10 Hz; process 2) and both are inversely related to the applied field strength. For NP6, NP8, and NP10, most of the high-frequency process 1 occurs outside the experimental range (>1500 Hz). Notable, process 1 is shifted for lower frequencies when decreasing the number of $Er^{3+}$ ions within the NP host, evidencing the impact of the paramagnetic ion isolation in decreasing the relaxation rate[48–52].

The field-dependent $\tau$ (relaxation time) values were obtained from the fit of the $\chi''$ susceptibility to the generalized double Debye model to consider both processes[53]. It should be noted the large distribution of $\tau$ values expressed as large α values (Tables S4–S9). The estimated standard deviations (ESD) of the relaxation time have been calculated from the α-parameter of the generalized Debye fits and the log-normal distribution as previously described[54]. This large distribution may occur due to inherent nature of ultrasmall nanoparticles in which the large surface/volume ratio results in a significant number of ions located close to the surface and possessing a higher degree of distortion. Upon careful inspection of $\tau$ values, it is evident that process 2 (for NP1-NP10) is predominately QTM (Fig. S5 and Table S10). Whereas for NP1-NP4, the $\tau$ values for the high-frequency process 1 were fit to Eq. (1), which includes the Raman, the field-dependent direct and tunneling mechanisms (Fig. S6 and Table S10):

$$\tau^{-1} = AH^4 + CT^n + \frac{B_1}{\left(1 + B_2H^2\right)} \qquad (1)$$

The best-fit parameters are summarized in Table S10. For process 1, the relaxation dynamics observed for $H_{dc} \leq 1000$ Oe are attributed to QTM, whereas when $H_{dc} \geq 2000$ Oe, the direct mechanism takes over as the predominant pathway. It should be noted that for process 1, QTM $B_1$ parameter values increase from NP1 to NP4. For process 1

QTM, it is not possible to unequivocally determine if the nature of the process is due to intraparticle, interparticle, or a combination of both. For example, in NP4 the mean distance between the $Er^{3+}$ ions was determined as 9.1 Å, while for NP2 the mean distance between the two $Er^{3+}$ ions was determined as 12.9 Å. If we assume two NP1 aggregated side-by-side (see TEM image, Fig. 1a), containing only one $Er^{3+}$ ion, and with the $Er^{3+}$ ion perfectly positioned in the center of the NP, the distance between these $Er^{3+}$ ions will be 26.9 Å. If the ions at two neighboring NPs are not located in the center but facing each other, the distance between the $Er^{3+}$ ions can be even smaller. At these distances, the presence of the process 1 QTM can be expected, and the lower $B_1$ value, when compared to NP2 and NP4, is in agreement with the increased distancing. Additionally, the observation of intraparticle-mediated QTM for NP1 cannot be fully ruled out. Although our data and calculations suggest the presence of only one $Er^{3+}$ per NP for NP1, NPs are not perfectly homogeneous in terms of composition, thus one can expect that a small but non-negligible amount of NPs contains more than one $Er^{3+}$ ion within the structure. Hence, unequivocally assignment of the process 1 QTM is not possible since both intra- and interparticle interactions can be reasonably explained based on the NP structural features. From this data, the optimal field to ensure minimal contributions from each of these mechanisms is 1800 Oe. At this field, the $\tau$ reaches the maximum values of 1.83, 0.74, 0.19 ms, and for NP1, NP2, and NP4, respectively. Such trends demonstrate that relaxation time at the atomic level for $Er^{3+}$ ion is highly influenced by the presence of nearest $Er^{3+}$ neighbors. Reducing the dopant concentration from 4 to 1 mol% (i.e., 4 $Er^{3+}$ ions per NP to one $Er^{3+}$ ion per NP), the large distribution of $\tau$ values became one order of magnitude slower at the optimal field (1800 Oe). This data suggests that SIM-like behavior attributed to $Er^{3+}$ ions can be attained at atomic level within a rigid NP matrix.

A closer inspection of low-frequency process 2 dynamics indicates an interparticle nature for this process because the QTM is present even at higher fields. Such behavior was previously observed for the $[Co_XZn_{1-X}(DAPBH)(NO_3)(H_2O)](NO_3)$ molecular system[55]. Furthermore, QTM relaxation rates for process 2 are in full agreement with dc data;

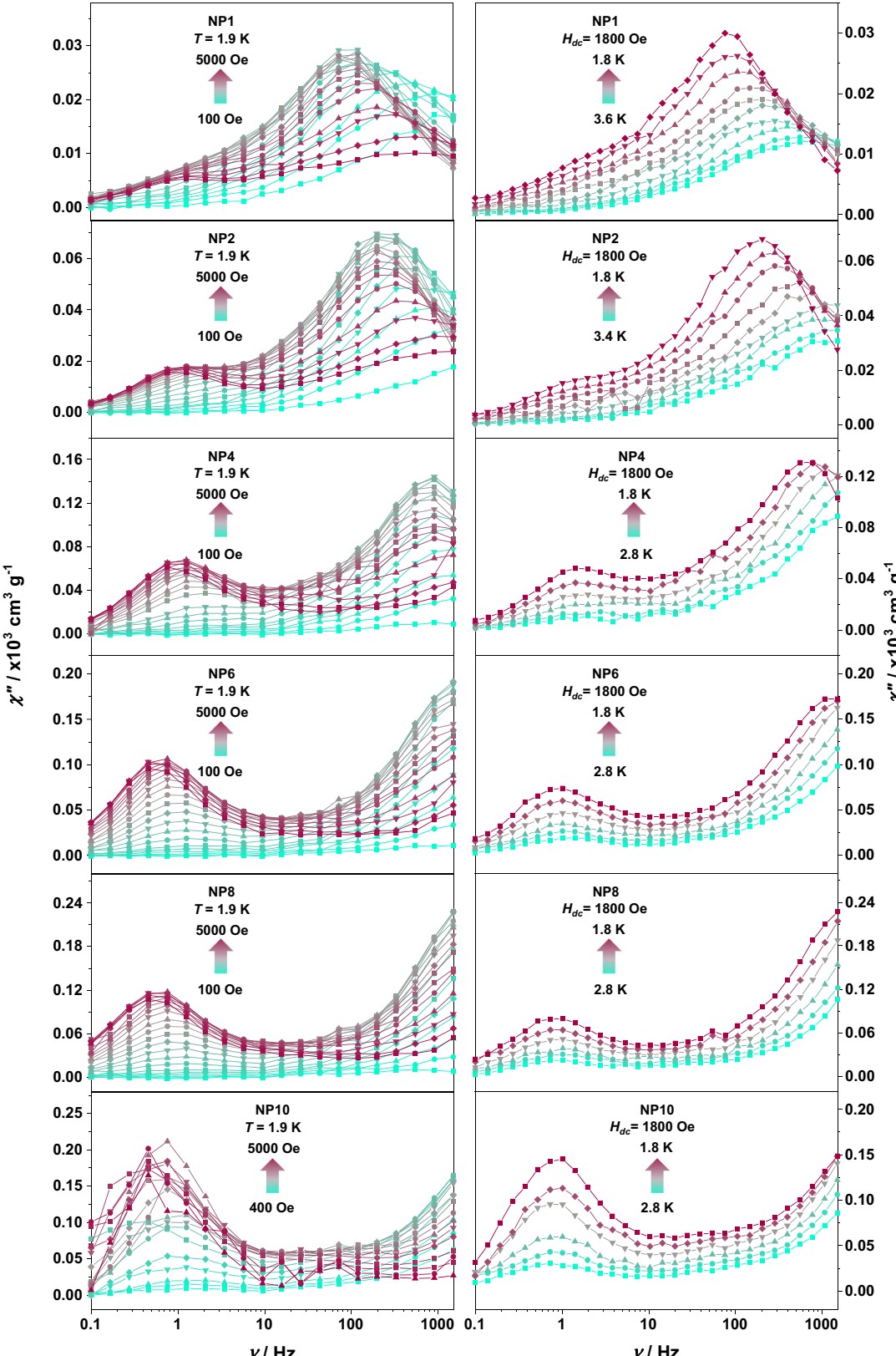

**Fig. 3 | Dynamic magnetic susceptibility data for the NPs.** Frequency dependence of the out-of-phase ($\chi''$) magnetic susceptibility for NP1 to NP10 (from top to bottom) obtained at 1.9 K as a function of the applied magnetic field (left) and under an applied static field of 1800 Oe as a function of temperature (right). Source data are provided as a Source Data file.

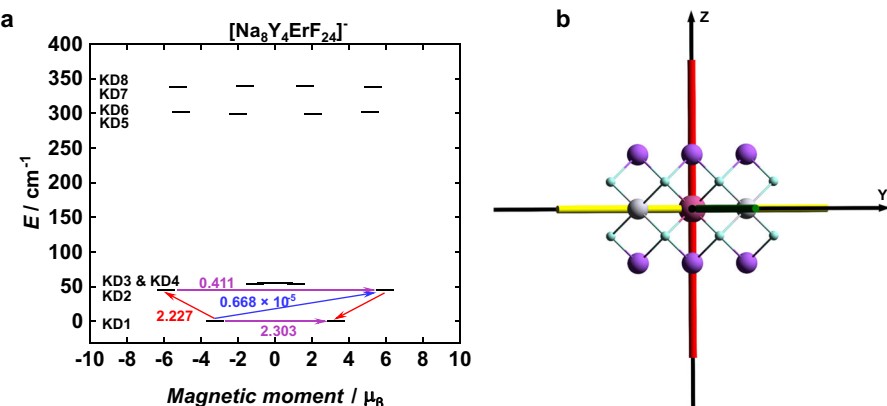

**Fig. 4 | Crystal field sublevels and magnetic axis for the $[Na_8Y_4ErF_{24}]^-$ theoretical model. a** Energy spectra and qualitative magnetization blocking barriers for $[Na_8Y_4ErF_{24}]^-$. Black lines represent the calculated Kramers doublets, and they are placed in the spectrum according to their magnetic moment. The numbers show the transition probabilities (in $\mu_B$) between the calculated Kramers doublets for the two lowest-lying spin-orbit states. Red, blue and purple lines correspond to direct, Orbach, and QTM processes, respectively. **b** Orientation of the main magnetic axes—$X_m$ (green), $Y_m$, (yellow) and $Z_m$ (red)—of the ground Kramers doublet of $Er^{3+}$ ion in $[Na_8Y_4ErF_{24}]^-$ fragment. See Fig. S9 for additional information and a comparison with other fragments.

for NP1, NP2, and NP4, $B_1$ values were above $10\ s^{-1}$ while for NP6, NP8, and NP10, the values decreased to below $10\ s^{-1}$, in agreement with the linearity deviation shown at Fig. 2b. This is in agreement with the expectations since not only intraparticle interactions are affected when more $Er^{3+}$ ions are confined within a single NP, but also the probability of interparticle interaction increases.

In order to further compare and probe the magnetic performance of the NPs, temperature-dependent relaxation dynamics were investigated under an external applied field of 1800 Oe (Fig. 3 right). Frequency-dependent out-of-phase ($\chi''$) susceptibilities were observed at low temperature, exhibiting magnet-like behavior, indicative of slow relaxation of the magnetization. As for the field-dependent relaxation dynamic, a bimodal profile was observed with similar behavior. The temperature-dependent $\tau$ values were obtained from the fit of the $\chi''$ susceptibility to the generalized double Debye model. As for the field-dependent data, process 2 is due to QTM originated by the interparticle dipolar interactions (Fig. S7 and Table S17). The $\tau$ values for process 1 were fit to Eq. (2):

$$\tau^{-1} = CT^n + \tau_{QTM}^{-1} \tag{2}$$

The best-fit parameters are summarized in Table S17. For NP4, NP2, and NP1, the Raman process alone is enough to reproduce the experimental data successfully. However, the relaxation rates became one order of magnitude slower when comparing NP4 and NP1 ($\tau_{1.8K}^{-1} = 3952$ and $478\ s^{-1}$ for NP4 and NP1, respectively). Such observation further affirms that a single $Er^{3+}$ ion confined within the NP matrix can act as a single-ion magnet. Attempts to obtain hysteresis measurements revealed no retention of the magnetic moment at 1.8 K for all the synthesized NPs, presumably due to the small energy barrier observed.

To investigate the microscopic magnetic properties of the synthesized NPs in detail, we carried out the CASSCF/SO-RASSI calculations for six different NP fragments—$[Na_8Y_4ErF_{22}]^+$, $[Na_8Y_4ErF_{23}]$, $[Na_8Y_4ErF_{24}]^-$, $[Na_8Y_{10}ErF_{40}]^+$, $[Na_8Y_{10}ErF_{41}]$, and $[Na_8Y_{10}ErF_{42}]^-$—with varying size, charge, and symmetry (see ESI for details). Comparison of the computational data to previously published results[56–58], particularly, to the EPR study of 8 nm cubic $NaY_{0.98}Er_{0.02}F_4$ nanoparticles[56], revealed that the $[Na_8Y_4ErF_{24}]^-$ fragment with $D_{4h}$ symmetry simulates the C.F. around the $Er^{3+}$ ion very well. Thus, only the results obtained for $[Na_8Y_4ErF_{24}]^-$ are discussed in detail (Fig. 4). Furthermore, to get insight on the phonon spectra of the investigated NPs, the periodic boundary conditions DFT (pDFT) calculations were carried out for $\alpha$-

phase of $NaYF_4$ and $NaY^{167.259}F_4$ of which the latter simulates the effects of $Er^{3+}$ ions on the phonon properties (see ESI for details).

An investigation of the weights of $M_J$ components reveals that the ground Kramers doublet (KD) of the $[Na_8Y_4ErF_{24}]^-$ fragment is composed of the $M_J = \pm13/2$, $\pm5/2$, $\pm3/2$, and $\pm11/2$ states, that is expected for the system in which $Er^{3+}$ ion occupies the $O_h$ site symmetry (Table S27; see ESI for details)[57]. The compositions of the excited KDs are also in line with the previously reported values for the $Ln^{3+}$ ions residing in the cubic environment and they indicate that the excited KDs cannot be either assigned to any pure $M_J$ states due to the pseudo $O_h$ site symmetry of the $Er^{3+}$ ion in the $[Na_8Y_4ErF_{24}]^-$ fragment strongly mix the different $M_J$ states[34]. As evident from the $g$-tensor ($g_x = 6.91$, $g_y = 6.91$, $g_z = 6.57$) of the ground KD of $[Na_8Y_4ErF_{24}]^-$, the calculations predict almost completely isotropic ground state for $Er^{3+}$ ions with the average $g$-value of 6.79 that is in excellent agreement with the $g$-value of 6.80 obtained from the EPR study of cubic $NaY_{0.98}Er_{0.02}F_4$ nanoparticle[56] (Table S24). Such a low axiality facilitates the QTM within the ground KD effectively, as proven by the calculated transition probability (Fig. 4). These results readily explain the fast relaxation of the magnetization of NPs without an applied dc field. If the QTM within the ground KD could be suppressed with the applied dc field, it is highly unlikely that the Orbach process *via* the low-lying first-excited KD ($44.8\ cm^{-1}$), as depicted in Fig. 4, would take place because the first excited KD has a larger magnetization than the isotropic ground KD. The result further supports the experimental findings that slow relaxation of magnetization originates from the Raman process and QTM in the studied NPs under an applied dc field. Interestingly, the DFT calculations reveal that the first optical phonon modes for $NaYF_4$ and $NaY^{167.259}F_4$ are at $125\ cm^{-1}$ and $122\ cm^{-1}$, respectively (Figs. S12 and S13). Given that the thermal population of optical phonon modes will be minimal in the low temperature region (<3.6 K), where NPs show slow relaxation of magnetization, it is more likely that acoustic phonons mediate the relaxation processes of the studied NPs. Optical phonons usually couple stronger to the spins than acoustic ones, but in the low temperature region the role of acoustic phonons cannot be ruled out[59]. In sharp contrast to molecular compounds, for which a high density of dispersive pseudoacoustic/low energy optical phonon modes are observed even below $20\ cm^{-1}$, these modes are absent in $NaYF_4$ and $NaY^{167.259}F_4$ (Figs. S12 and S13)[15,60]. As proposed by Lunghi et al., the high frequency value of the first optical phonon mode of the crystal cell, as observed for $NaYF_4$ and $NaY^{167.259}F_4$, could be taken as one of the design strategies to ensure the small amount of

low-energy vibrations in the lattice, low phonon density of states at the spin resonance energies, and rigidity of ligands to minimize the coupling of intramolecular motions to low-energy acoustic phonons[15]. Indeed, $NaYF_4$ and $NaY^{167.259}F_4$ have a small amount of low-energy vibrations (Fig. S10 and Tables S21 and S20) and the energies of the first four KDs of the $[Na_8Y_4ErF_{24}]^-$ fragment are in the region (0–54 cm$^{-1}$) where the phonon density of states is small (Table S24 and Figs. S12 and S13). Despite these facts, low-energy acoustic phonons most likely couple to the spins of $Er^{3+}$ centers because slow relaxation of magnetization of the investigated NPs takes place in the temperature region where only the thermal population of acoustic phonons is significant, and the energies of the higher lying KDs (KD5–KD8) are in an energy range of 300–350 cm$^{-1}$ of optical phonons with a high density of states. The former can be partly explained by the fact that a large part of phonon distributions originates from F and Er atoms in the low-energy region of acoustic phonons, as shown by the projected density of states (Fig. S13). While the phonon density of states does not always correlate with the spin-phonon coupling intensities[15], the above qualitative analysis shows that the $NaYF_4$ NPs can function as host materials that partly obey the current design principles for the high-performing SIMs.

The calculated C.F. parameters ($B_{kq}$) show an interesting trend for $[Na_8Y_4ErF_{24}]^-$. The $B_{40}$ and $B_{60}$ diagonal parameters contribute significantly to the C.F. However, the off-diagonal parameters $B_{44}$, and $B_{64}$ also substantially contribute to reducing the overall axiality of the fragment (Table S33). All other C.F. parameters are essentially zero for $[Na_8Y_4ErF_{24}]^-$. These features originate from the $O_h$ site symmetry of the doped NPs. Thus, the CASSCF/SO-RASSI calculations do not only indicate the modest SIM behavior of investigated $Er^{3+}$-doped NPs, but they also highlight that SIM based on NPs doped at low concentration can strictly follow the symmetry design criteria of SMM[61]. This is a significant finding because it is not easy to synthesizes the molecular system that would possess an exact high symmetry coordination environment around $Ln^{3+}$ ion obeying the symmetry design criteria rigorously. However, ideal local $O_h$ symmetry around the $RE^{3+}$ ion will automatically lead to a system in which the ground KD is completely isotropic, like in $[Na_8Y_4ErF_{24}]^-$ (Table S24), and prone to strong QTM[32]. Thus, future synthetic efforts should focus on doped NPs in which the high crystal symmetry is accompanied by a high local symmetry other than $O_h$ symmetry. For example[62], in doped fluorites, the $Er^{3+}$ ions can form a hexameric cluster with $F^-$ ions in which the $Er^{3+}$ ions occupy the $D_{4d}$ site symmetry that is known to stabilize the axial ground KD ($g_x \approx g_y \approx 0.00$, $g_z \approx 18.00$) with almost pure $M_J \pm 15/2$ state in case of $Er^{3+}$ ions[34].

In summary, by applying the knowledge accumulated over the last decade in the molecular field, we successfully demonstrated that slow magnetic relaxation could be attained for a single $Er^{3+}$ ion within a nanoparticle matrix. This was achieved by implementing doping into a rigid diamagnetic $\alpha$-$NaYF_4$ NP matrix. Upon dilution at 1 mol% concentration, it is possible to isolate a single paramagnetic $Er^{3+}$ ion that sits in a cubic coordination environment ($O_h$ symmetry) in these ultrasmall sub-3 nm NPs. At this low concentration, $Er^{3+}$-$Er^{3+}$ interactions were minimized, which resulted in slower relaxation rates when compared with NPs containing more than one $Er^{3+}$ ion. Although the relaxation process occurs via Raman and QTM processes in the low temperature region where acoustic phonons dominate the phonon spectra, this study proves that this methodology can be extended to NPs with greater crystal symmetry and rigid framework with further enhanced SMM behavior. The employed concentration dependant study further demonstrates that interatomic distances can be controlled to correlate various relaxation mechanisms promoted due to proximity between paramagnetic centers. We envision such studies can ultimately enable careful and selective doping of traditional magnets and other magnetic materials to isolate the next generation of ultra-hard magnets.

## Methods

### Nanoparticle synthesis

Yttrium oxide ($Y_2O_3$, 99.99%) and erbium oxide ($Er_2O_3$, 99.99%) were purchased from Alfa Aesar. Oleic acid ($CH_3(CH_2)_7CHCH(CH_2)_7COOH$, OA, 90%), oleylamine ($CH_3(CH_2)_7CHCH(CH_2)_7CH_2NH_2$, OAm, 70%), 1-octadecene (ODE, 90%), and ammonium fluoride ($NH_4F$, 98%) were purchased from Sigma-Aldrich. Sodium oleate ($CH_3(CH_2)_7CHCH(CH_2)_7COONa$, Na-OA, 97%) was purchased from Tokyo Chemical Industry. Toluene (99.8%) was purchased from Fisher Scientific. Ethanol (99%) and hexane (analytical grade) were purchased from Commercial Alcohols and Fisher Chemicals, respectively. All chemicals were used as received.

Cubic-phase $Er^{3+}$-doped $NaYF_4$ nanoparticles were synthesized using the microwave-assisted thermal decomposition of $RE^{3+}$ oleate precursors ($[RE(OA)_3]$, OA: $CH_3(CH_2)_7CH = CH(CH_2)_7COO^-$), as previously reported[38]. $[RE(OA)_3]$ precursors were prepared following an adopted procedure reported in the literature[63]. In brief, 0.625 mmol of $RECl_3 \cdot 6H_2O$ (RE = $Y^{3+}$ and $Er^{3+}$—the relative amounts of chlorides for each NP composition are given in Table S1), 3.75 mmol (1141.7 mg) of sodium oleate (Na-OA), and a mixture of 2.8 ml of water, 3.8 ml of ethanol, and 6.6 ml of hexane were added to a 50 ml round-bottom flask. The flask was fitted to a condenser and the mixture was refluxed under vigorous stirring for 2 h at 70 °C. Subsequently, the organic layer containing $[RE(OA)_3]$ was washed three times with 4 ml of a 1:1 water-to-ethanol mixture in a separatory funnel. Ten ml of 1-octadecene (ODE) was added to the $[RE(OA)_3]$ solution in hexane and the residual solvent was evaporated at 70 °C under a gentle flow of $N_2$ for 20 min. For the synthesis of $\alpha$-$NaYF_4$:$Er^{3+}$ NPs ($Er^{3+}$ dopant concentration X = 1, 2, 4, 6, 8 or 10 mol%), 1.15 mmol (380.55 mg) of Na-OA and 10 ml of oleic acid were added to the $[RE(OA)_3]$-ODE mixture (resulting in a 1:1 $Na^+$-to-$RE^{3+}$ molar ratio), and the solution was degassed under vacuum at 100 °C for 30 min. A 10 ml aliquot of this reaction mixture was transferred into a 35 ml microwave vessel containing 2.5 mmol (94.5 mg) of $NH_4F$, purged with $N_2$, tightly sealed, inserted into a CEM Discover SP microwave, and subjected to the following reaction profile: (1) medium stirring at room temperature for 1 min and rapid heating to 100 °C (1 min) under slow stirring, (2) slow stirring with rapid increase in temperature to 240 °C (5 min), (3) rapid cooling to 230 °C (15 s), (4) static heating at 230 °C (10 min) under slow stirring, and (5) gradual cooling to 50 °C (6 min). Following the synthesis, the reaction mixture containing the NPs was transferred into a centrifugation tube, diluted with a 1:3 hexane-to-ethanol mixture and centrifuged at 6595 × g for 20 min. The product was then washed with a 1:3 toluene-to-acetone mixture and centrifuged using the same conditions reported above. After purification, the NPs were dispersed and stored in 5 ml hexane for further use.

Samples for TGA and SQUID measurements were dried overnight in a Schlenk line.

### Characterizations and methods

To determine the crystal phase of the NPs, powder X-ray diffraction (XRD) analysis was performed using a Rigaku Ultima IV Diffractometer (Cu K$\alpha$, $\lambda$ = 1.5401 Å), operating at 44 kV and 40 mA (step size: 0.02°, scan speed: 0.7° min$^{-1}$). The size and morphology of the NPs were determined by transmission electron microscopy (TEM) using a FEI Tecnai Spirit microscope operating at 120 kV. The samples were diluted in hexane from the original stock suspensions by 50 times and 6 μl of the diluted suspension was dropped on a Formvar/carbon film supported by a 300-mesh copper TEM grid. Thermogravimetric analysis (TGA) of the NPs was carried out using a TGA Q500/Discovery under $N_2$ atmosphere and at a heating rate of 10 °C min$^{-1}$.

The $Er^{3+}$ dopant concentration in each of the NPs compositions was determined by inductively coupled plasma–optical spectrometry (ICP-OES) using an Agilent ICP-OES spectrometer (nebulizer flow of 0.7 l min$^{-1}$, plasma flow 12 l min$^{-1}$, auxiliary flow 1 l min$^{-1}$). The samples

were prepared by digesting 10 μl of the NP stock suspensions in hexane with 0.5 ml of $HNO_3$ and 1.5 ml of HCl at 40 °C overnight, followed by the dilution with distilled water to 5 ml.

The magnetic susceptibility measurements were obtained using a Quantum Design SQUID magnetometer MPMS-XL7 operating between 1.8 and 300 K. Direct (dc) and/or alternate (ac) current measurements were performed on dried samples restrained with silicon grease and wrapped in a polyethylene membrane. The samples were subjected to dc fields of −7 to 7T, and a 3.78 Oe driving field was used for ac measurements. The magnetization data were collected at 100 K to check for ferromagnetic impurities that were absent in all samples. Diamagnetic corrections were applied for the sample holder and the inherent diamagnetism of the samples were estimated with the use of Pascals constants.

## Data availability

All data obtained and analyzed are present in the manuscript and the Supplementary Material. The raw data for Figs. 2 and 3 are made available as source data. All the other raw data that support the findings of this study are available from the corresponding authors upon request. Source data are provided with this paper.

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

## Acknowledgements

D.A.G. and M.M. thank the University of Ottawa, the CFI, the Natural Sciences and Engineering Research Council of Canada for financial support of this work. E.H., E.M.R. and I.H. gratefully acknowledge the financial support provided by the University of Ottawa and the Natural Sciences and Engineering Research Council of Canada (NSERC, RGPIN-2016-04830). J.T. and J.O.M. acknowledge the Research council of Finland (projects 315829, 345484, and 338733) for the financial support, and Prof. Heikki M. Tuononen (University of Jyväskylä) for providing computational resources for the project. We also thank Prof. Antti Karttunen for the useful discussion related to the periodic boundary density functional theory calculations.

## Author contributions

D.A.G., E.H. and M.M. conceived the study. I.H., E.M.R. and E.H. synthesized and characterized the nanoparticles. D.A.G. collected and interpreted the magnetic data. H.Z., J.X. and X.L. performed Monte Carlo simulations and analysis. J.T. and J.O.M. performed theoretical calculations and analysis. M.M. supervised all aspects of the project. The manuscript was written with contributions from all authors.

## Competing interests

The authors declare no competing interests.
