## [Peer Review File · Nature Communications]

Reviewers' Comments:

Reviewer #1:

Remarks to the Author:

This is a manuscript I previously reviewed for Nature Chemistry. It reports on the magnetic properties of NaYF₄ nanoparticles doped with controlled amount of Er(III) ions. The degree of synthetic control that authors obtain is remarkable and the study is quite complete, however both the results of the magnetic properties analysis and the description of the general purpose of the study require major revisions before acceptance in Nature Communications can be suggested. In this revised version, authors included a few specific modifications following remarks of both mine and other reviewers' reports. However, they seemed willing to keep the changes in the text to a minimum, while answering in some more detail in their rebuttal letter. In particular, they answered to my first three remarks by using essentially the same consideration, i.e. that the underlying idea of the paper is that of demonstrating that "it is possible to bring the knowledge acquired in molecular system to the nanoparticle field and show it is possible to observe slow-relaxation dynamics in nanoparticles when dilution is carefully controlled". It is much probably my fault, but by only reading the manuscript I did not get the impression that this was the scope of the study, which in the abstract is described in somehow different terms ("To understand the role and the importance of molecular rigidity, and probe the slow relaxation dynamics intrinsic to the single Er³⁺ ion"). In perspective, they suggest that "The knowledge gained from this study is anticipated to enable better design of magnetically high performing molecular and bulk magnets for a wide variety of applications...". From my point of view, here a rewriting of the introductory part by making clearer their point for a general public.

As for the remaining and more specific objections I raised, I find that authors should find a way to include part of the answers they provided in the letter in the manuscript, which remains unclear in many points and requires several corrections and major revisions, as detailed in the following:

- In my previous report (but see also those of the other reviewers) I raised some concerns on the choice of the system, which I did not find the optimum one to maximize the anisotropy. Authors answered in some detail in their rebuttal letter, but left the text essentially unchanged, stating that "among all RE³⁺ ions, Er³⁺ is ideally suited to benefit from the cubic crystal field to optimize the magnetic anisotropy". This, as already pointed out and as recognized by the authors themselves in their letter, is questionable: there is no "benefit" toward optimization of magnetic anisotropy (of the axial type required for slow relaxation, at least) with cubic systems. Rather, in their answer in the rebuttal letter, authors suggest that in a context of distorted cubic symmetry, which tends to quench the axial magnetic anisotropy, Er(III) can be considered to suffer less than other lanthanide ions. In its present version the sentence in the manuscript is misleading for the general public to which Nature Communications is addressed and has then to be changed.

- Figure 2 now correctly reports the temperature dependent magnetic dc properties as $M \cdot T / H$ as required by two reviewers: however, the text apparently remained unchanged, and still mentions "linear magnetization value". This is not what the graphs show, since essentially constant MT values are observed: thus M is not linear in T , but rather in $1/T$. The discussion reported in the rebuttal letter about the difficulties in rescaling vs the moles of Er(III) ions should also be somehow included in the manuscript. Furthermore, the field at which the measurements were performed must be added in the Experimental Section. I notice here that, if the observed different behavior with increasing Er(III) content is to be attributed to dipolar interactions, measurements at high and low field should evidence this, since the estimated dipolar interaction, which is relatively small, should be overcome by large fields. If the authors have the possibility to measure a few of their samples in different magnetic fields, I strongly advise them to do this.

- The point about the overparametrization of the fit of field and temperature dependence of relaxation rate needs a clarification, since apparently my previous remark was not clear enough. While it is obvious that QTM can be fixed to a constant when analyzing the temperature dependence of the relaxation rate, this value should be ideally equal to $B_1 / (1 + B_2 H^2)$. Any deviation from this value indicates some inconsistency in the model. The same consideration applies for the value of the parameters describing the Raman contribution in both field and temperature dependencies. This is quite a standard procedure and should be followed in a paper aimed at Nature Communications.

- One of the main modifications included by the authors compared to the original version concerns their answer to the suggestion by reviewer #4 to "calculate the dipolar exchange interactions and correlate with experiment". It is my understanding that they did nothing of the sort, and just

included a discussion using the results on a previously reported organometallic complex to prove that 8 Angstrom is the limiting distance to have an effect of the dipolar interaction. However, by having the calculated g values, their orientation, the average distance between Er(III) ions (and even their distribution), the minimum one can expect in a Nature Communications paper is to have the dipolar interactions calculated and correlated with magnetic data, since this is one of the main point of the paper. Without these calculations it is my opinion that the manuscript cannot be accepted for publication in Nature Communications.

A few further minor points are detailed below:

- In Supplementary Material, please include reference to Iwahara- Chibotaru paper for completeness sake.

- on l.289/290, they identify $q = 1, 3, 5$, as "non-axial"; I find this potentially misleading, since the axis to which one is referring has not yet been defined, and axial terms are usually considered only those with $q=0$.

- A minor point, which authors may wish to include in the discussion, concern the role of the hyperfine coupling. Er has both zero and non-zero nuclear spins isotopes in non-negligible natural abundance. Do they think this might explain the two different processes? A comment on this point would be welcome.

Reviewer #2:

Remarks to the Author:

Gálico et al. have presented a revised manuscript based on the supposed isolation of single or few Er³⁺ ion in nanoparticles. There are significant issues in presentation and justification that, to me, need to be addressed before publication. I think synthetically that these results are interesting, but that the characterisation does not currently support their explanation of the data.

lines 62-64: "These molecular vibrations can cause spin-phonon relaxation which leads to the reduction of the energy barrier (U_{eff}) and the overall magnetic performance of a system." - this is factually inaccurate. Spin phonon coupling allows magnetic relaxation, which may occur via the Orbach mechanism which leads to observation of an effective energy barrier U_{eff} . The U_{eff} may be lower than the total splitting of the low-lying manifold of states, but this is not the fault of the phonons, but rather how the phonons couple the mixed electronic states.

lines 118-120: "The high symmetry cubic coordination environment provided by α -NaYF₄ favours the enhancement of the magnetic anisotropy rather than the nine-coordinate environment in the β -phase." - again, this is not true. Oh symmetry is, by definition, higher than any nine-coordinate geometry, and thus cannot "favour the enhancement of the magnetic anisotropy".

lines 121-122: "Among all RE³⁺ ions, Er³⁺ is ideally suited to benefit from the cubic crystal field to optimize the magnetic anisotropy." - based on what? This statement is unjustified, and I am quite sure it cannot be true. The limiting anisotropic shapes of f-electron density for the maximal and minimal mJ projections for Ln(III) ions are prolate (elongated) and oblate (compressed) spherioids - Oh does not favour either one of these. As such Oh symmetry does not optimize magnetic anisotropy for any Ln(III) ion, and rather minimises it. See previous comment. Hence, the sentence "Thus, Er³⁺ is anticipated to perform better by exhibiting slow relaxation of the magnetisation than other RE³⁺ ions.¹⁹" to me, is also untrue - it seems that in that reference those authors make some fundamentally incorrect statements.

Lines 165-166: "The increase of the magnetisation values below this temperature is likely due to a combination of thermal depopulation of the Stark sublevels" - this should say "decrease", not "increase", but also I suspect the larger decrease for the higher concentrations relative to the lower concentrations is due to stronger magnetic interactions within the NP. I note that the high temperature data, especially for the most dilute samples, decreases linearly with increasing temperature - have these data been corrected for diamagnetism?

Fig 3: Why are the AC data the noisiest for sample NP10, which has the largest magnetic concentration?

Figs S5-S7: The underlying Debye fits show very large alpha values, corresponding to large distributions larger than one order of magnitude in some cases. See Reta and Chilton (10.1039/C9CP04301B). Ranges corresponding to these distributions should be indicated on the plots, and discussed in the main text. I don't disagree with fitting the central tau values to obtain parameters, but it must be emphasised that these parameters are the central profiles of incredibly large distributions across the samples. Thus, statistically, I am not sure one can extract reliable information on "single" Er³⁺ sites - simply because there is not one definition of such a site, according to their magnetic data. Eg, the lines 228-230: "Reducing the dopant concentration from 4 to 1 mol% (i.e., 4 Er³⁺ ions per NP to one Er³⁺ ion per NP), a nearly 10-fold enhancement can be attained for the magnetisation relaxation times. This data further validates that SMM-like behaviour attributed to one single Er³⁺ ion can be attained at atomic level within a rigid NP matrix" are not a faithful description of the data: there is more than 100-fold variation of relaxation times within each sample, so statements like this are far too strong, and need to be appropriately tempered.

Fig S7: I am not convinced of the need to introduce an Orbach fitting parameter for the NP1(P1) data. These P1 data should be shown on a smaller scale. However, given the large distributions in relaxation times (due to large alpha), and significant variation in CASSCF data, I do not think there is strong justification here.

Table S4-S16: Some values given in scientific notation and some not - please be consistent.

Lines 235-238: "Although the data suggest that the origin of the QTM process is due to the intraparticle Er³⁺-Er³⁺ interaction, it is not possible to completely rule out the presence of a small but non-negligible interparticle Er³⁺-Er³⁺ interaction, as even for NP1, a small contribution in the low-frequency QTM process is still present" - examination of the fitted B1 values, which give the contribution of the QTM process, show that QTM is apparently more significant for NP1 (16e3) than for NP4 (11e3), so this argument does not hold water. Moreover, even if this statement was true, then it rather disagrees with their previous assertion that the NP1 samples contain single Er³⁺ ions. Things seem to look more sensible in the temperature-dependant data (lines 239-243, Table S17), however these data are collected at 1800 Oe where QTM should largely suppressed. Indeed, it is not clear to me that the interpretation of the data in Fig S5 are indeed QTM - this is apparently occurring over a vastly larger field range than commonly observed for QTM (usually < 1000 Oe, as seen in Fig S6).

Tables S18-24: First point: Tables S21 and S23 shows all states have no symmetry, and yet all the relative angles between "main" magnetic axes are 90.00 - this is not sensible, and the data should be checked. Overall, the calculations show a significant change in the character of the ground KD of the Er³⁺ site depending on the model. While the authors attempt to justify their discussion of the [Na₅Y₈ErF₃₂] model owing to the "convergence" of the results for the larger cluster calculations to a mJ ± 15/2 ground state, the carefully omit the [Na₆Y₄ErF₂₄]₃- model from their statement, which is a large model that shows a mJ ± 1/2 ground state. Further, it is the only model that approaches the true site symmetry of the compound: I am not convinced that a true high symmetry site in this compound should be able to stabilise the ca. 90% ± 15/2 ground state they claim. Clearly, symmetry AND size AND charge are important here, and things change a lot, apparently. As the magnetic data do not uniquely identify one ground state over another, I do not see justification for one choice over another. One thing the authors should check is that the quantization axes for the CFP and basis definitions is the same in all cases. A robust way to prove this, one way or the other, is using EPR spectroscopy - a low temperature CW X- or Q-band spectrum should be very diagnostic here, and greatly improve the manuscript by evidencing the magnetic ground state.

Tables S25-31: One cannot define a percentage of real and imaginary components. The contribution of a given basis function to the wavefunction is conj(Ci)*Ci where Ci is the complex coefficient and conj(Ci) its complex conjugate. The sum of all conj(Ci)*Ci for a given state will be 1: this is the normalisation condition, and thus can be represented as a percentage.

REVIEWER COMMENTS

Reviewer #3 (Remarks to the Author):

This is a manuscript I previously reviewed for Nature Chemistry. It reports on the magnetic properties of NaYF₄ nanoparticles doped with controlled amount of Er(III) ions. The degree of synthetic control that authors obtain is remarkable and the study is quite complete, however both the results of the magnetic properties analysis and the description of the general purpose of the study require major revisions before acceptance in Nature Communications can be suggested.

We thank the reviewer for evaluating our manuscript and giving us another round of constructive feedback to improve our manuscript.

In this revised version, authors included a few specific modifications following remarks of both mine and other reviewers' reports. However, they seemed willing to keep the changes in the text to a minimum, while answering in some more detail in their rebuttal letter. In particular, they answered to my first three remarks by using essentially the same consideration, i.e. that the underlying idea of the paper is that of demonstrating that "it is possible to bring the knowledge acquired in molecular system to the nanoparticle field and show it is possible to observe slow-relaxation dynamics in nanoparticles when dilution is carefully controlled". It is much probably my fault, but by only reading the manuscript I did not get the impression that this was the scope of the study, which in the abstract is described in somehow different terms ("To understand the role and the importance of molecular rigidity, and probe the slow relaxation dynamics intrinsic to the single Er³⁺ ion"). In perspective, they suggest that "The knowledge gained from this study is anticipated to enable better design of magnetically high performing molecular and bulk magnets for a wide variety of applications....". From my point of view, here a rewriting of the introductory part by making clearer their point for a general public.

We thank the reviewer for bringing up this specific point. We agree that the message in the intro was not clear. To improve, we have rewritten some parts of the abstract, introduction, and conclusion to bring a more accurate picture of the main goals of this study.

As for the remaining and more specific objections I raised, I find that authors should find a way to include part of the answers they provided in the letter in the manuscript, which remains unclear in many points and requires several corrections and major revisions, as detailed in the following:

- In my previous report (but see also those of the other reviewers) I raised some concerns on the choice of the system, which I did not find the optimum one to maximize the anisotropy. Authors answered in some detail in their rebuttal letter, but left the text essentially unchanged, stating that “among all RE³⁺ ions, Er³⁺ is ideally suited to benefit from the cubic crystal field to optimize the magnetic anisotropy”. This, as already pointed out and as recognized by the authors themselves in their letter, is questionable: there is no “benefit” toward optimization of magnetic anisotropy (of the axial type required for slow relaxation, at least) with cubic systems. Rather, in their answer in the rebuttal letter, authors suggest that in a context of distorted cubic symmetry, which tends to quench the axial magnetic anisotropy, Er(III) can be considered to suffer less than other lanthanide ions. In its present version the sentence in the manuscript is misleading for the general public to which Nature Communications is addressed and has then to be changed.

We thank the reviewer for this comment. As for the previous comment, we changed part of the manuscript to reflect the discussions about these comments/answers. Most of the points answered in the previous round were now incorporated into the manuscript.

- Figure 2 now correctly reports the temperature dependent magnetic dc properties as $M \cdot T/H$ as required by two reviewers: however, the text apparently remained unchanged, and still mentions “linear magnetization value”. This is not what the graphs show, since essentially constant MT values are observed: thus M is not linear in T , but rather in $1/T$. The discussion reported in the rebuttal letter about the difficulties in rescaling vs the moles of Er(III) ions should also be somehow included in the manuscript. Furthermore, the field at which the measurements were performed must be added in the Experimental Section. I notice here that, if the observed different behavior with increasing Er(III) content is to be attributed to dipolar interactions, measurements at high and low field should evidence this, since the estimated dipolar interaction, which is relatively small,

should be overcome by large fields. If the authors have the possibility to measure a few of their samples in different magnetic fields, I strongly advise them to do this.

We thank the reviewer for this comment. We changed the word "linear" to "constant" and added the field in the figure.

We also added the 10000 Oe field data (Figures 2 b and d). The data corroborates our discussion. As can be observed, the magnetization value at 1.8 K for different compositions now follows linearity up to NP8, confirming that the deviation previously observed is, in fact, due to intraparticle dipolar magnetic interactions.

- The point about the overparametrization of the fit of field and temperature dependence of relaxation rate needs a clarification, since apparently my previous remark was not clear enough. While it is obvious that QTM can be fixed to a constant when analyzing the temperature dependence of the relaxation rate, this value should be ideally equal to $B_1/(1+B_2H^2)$. Any deviation from this value indicates some inconsistency in the model. The same consideration applies for the value of the parameters describing the Raman contribution in both field and temperature dependencies. This is quite a standard procedure and should be followed in a paper aimed at Nature Communications.

Following the reviewer's recommendation, we reanalyzed the field-dependent data, and now the field and temperature-dependent parameters are consistent between them (Tables S10 and S17). It should be noted that our previous set of parameters, although not perfectly matching, were close, always in the same order of magnitude. Therefore, the provided discussions are not influenced by the new set of parameters. I hope this addresses the reviewer's concern.

- One of the main modifications included by the authors compared to the original version concerns their answer to the suggestion by reviewer #4 to "calculate the dipolar exchange interactions and correlate with experiment". It is my understanding that they did nothing of the sort and just included a discussion using the results on a previously reported organometallic complex to prove that 8 Angstrom is the limiting distance to have an effect of the dipolar interaction. However, by having the calculated g values, their orientation, the average distance between Er(III) ions (and

even their distribution), the minimum one can expect in a Nature Communications paper is to have the dipolar interactions calculated and correlated with magnetic data, since this is one of the main point of the paper. Without these calculations it is my opinion that the manuscript cannot be accepted for publication in Nature Communications.

We have now calculated the dipolar coupling parameters between two interacting Er^{3+} ions that are separated from each other by the distance of r in four different distances of 3.868 Å, 5.470 Å, 8.649 Å, and 10.940 Å. The following values were obtained for the dipolar coupling parameters: $J_{dip} = -1.07 \text{ cm}^{-1}$ ($r = 3.868 \text{ Å}$), $J_{dip} = -0.38$ ($r = 5.470 \text{ Å}$), $J_{dip} = -0.09$ ($r = 8.649 \text{ Å}$), and $J_{dip} = -0.05$ ($r = 10.940 \text{ Å}$). The results show that the strength of the dipolar coupling decreases with the increasing $\text{Er}^{3+}\cdots\text{Er}^{3+}$ distance and the coupling at $\sim 8 \text{ Å}$ is two orders of the magnitude smaller than at $\sim 3 \text{ Å}$. The results are in line with experimental data, and they are now incorporated in the manuscript.

A few further minor points are detailed below:

- In Supplementary Material, please include reference to Iwahara- Chibotaru paper for completeness sake.

Thank you for pointing these appropriate references out. They are now included in the ESI.

- on 1.289/290, they identify $q = 1, 3, 5$, as “non-axial”; I find this potentially misleading, since the axis to which one is referring has not yet been defined, and axial terms are usually considered only those with $q=0$.

We apologize for using this misleading term. We have now changed the terms “non-axial” and “axial” to “non-diagonal” and “diagonal”, respectively, when discussing the CFPs.

- A minor point, which authors may wish to include in the discussion, concern the role of the hyperfine coupling. Er has both zero and non-zero nuclear spins isotopes in non-negligible natural abundance. Do they think this might explain the two different processes? A comment on this point would be welcome.

We thank the reviewer for the comment. We acknowledge that the hyperfine coupling mediated by the presence of different isotopes can impact the magnetic properties of SIMs. This fact was already widely demonstrated for dysprosium-based SMMs (10.1039/C7CC00317J, 10.1039/C8QI01209A, 10.1002/ejic.201700842, and 10.1002/ange.201409887) and in fewer examples for ytterbium (10.1021/acs.inorgchem.0c02652) and erbium (10.1002/chem.202100953) based SMMs. Although we recognize that the presence of different isotopes may impact the magnetic relaxation, we do not see any indication in our work that this may be the reason for the presence of two different processes. Both processes can be explained by inter- and intra-particle interactions and by the crystal field around the erbium ion. We do not think that, based on our results, we could be able to hypothesize a discussion based on hyperfine coupling without isotopic enrichment experiments. We agree that the present suggestion would be very interesting; however, it is outside the main focus of this work, and the presented discussion seems enough to explain the SIM behaviour of the erbium ion confined in the nanoparticle host.

Reviewer #4 (Remarks to the Author):

Gálico et al. have presented a revised manuscript based on the supposed isolation of single or few Er³⁺ ion in nanoparticles. There are significant issues in presentation and justification that, to me, need to be addressed before publication. I think synthetically that these results are interesting, but that the characterisation does not currently support their explanation of the data.

We thank the reviewer for the comment and following suggestions.

lines 62-64: "These molecular vibrations can cause spin-phonon relaxation which leads to the reduction of the energy barrier (U_{eff}) and the overall magnetic performance of a system." - this is factually inaccurate. Spin phonon coupling allows magnetic relaxation, which may occur via the Orbach mechanism which leads to observation of an effective energy barrier U_{eff} . The U_{eff} may be lower than the total splitting of the low-lying manifold of states, but this is not the fault of the phonons, but rather how the phonons couple the mixed electronic states.

We thank the reviewer for this specific point. We reworded the sentence to: “*The high density of molecular vibrations increases the probability of the phonons coupling with electronic states, resulting in phonon-assisted relaxation which leads to the reduction of the energy barrier (U_{eff}) and the overall magnetic performance of a system*”. The reworded sentence now focuses on the fact that molecular compounds possess a higher density of vibrational states when compared with nanoparticles; thus, the probability of coupling between phonons and electronic states is higher for molecular compounds.

lines 118-120: "The high symmetry cubic coordination environment provided by α -NaYF₄ favours the enhancement of the magnetic anisotropy rather than the nine-coordinate environment in the β -phase." - again, this is not true. Oh symmetry is, by definition, higher than any nine-coordinate geometry, and thus cannot "favour the enhancement of the magnetic anisotropy".

lines 121-122: "Among all RE³⁺ ions, Er³⁺ is ideally suited to benefit from the cubic crystal field to optimize the magnetic anisotropy." - based on what? This statement is unjustified, and I am quite sure it cannot be true. The limiting anisotropic shapes of f-electron density for the maximal and minimal mJ projections for Ln(III) ions are prolate (elongated) and oblate (compressed) spherioids - Oh does not favour either one of these. As such Oh symmetry does not optimize magnetic anisotropy for any Ln(III) ion, and rather minimises it. See previous comment. Hence, the sentence "Thus, Er³⁺ is anticipated to perform better by exhibiting slow relaxation of the magnetisation than other RE³⁺ ions.¹⁹" to me, is also untrue - it seems that in that reference those authors make some fundamentally incorrect statements.

We agree that the perfect O_h site symmetry does not provide an optimal crystal field around Ln³⁺ ions as it stabilizes the isotropic ground state instead of the anisotropic one. However, any minor distortion from the O_h site symmetry can significantly increase the magnetic anisotropy of Er³⁺ ion; for example, see 10.1021/ic302068c. Thus, we have reworded the introduction and explained the limitations of O_h symmetry to increase the magnetic anisotropy of Er³⁺ ions:

“Although the pure O_h symmetry stabilizes the isotropic ground state that is prone to the fast relaxation of the magnetization contrast to the anisotropic ground state,²⁰ any distortions on the O_h site can increase the magnetic anisotropy of Er³⁺ ions.²¹ With that said, some small but non-

negligible distortion around the RE³⁺ coordination environment is expected given the different ionic radii of the dopant and host RE³⁺ ion (1.15 Å for Y³⁺ and 1.14 Å for Er³⁺).^{22,23}

We also agree that the nine-coordinated environment in the β -phase could be better for enhancing the magnetic anisotropy than the eight-coordinated cubic environment in the α -phase. Thus, we have now reworded the introduction and clarified the selection of the α -phase over the β -phase in terms of the simplicity of the α -phase:

“In theory, the single high symmetry cubic coordination environment provided by α -NaYF₄ should eliminate the multiple relaxation processes originating from the different site symmetries of occupied sites that could be observed for β -phase due to the two different nine-coordinate C_{3v} sites. In other words, α -phase is more straightforward to investigate than β -phase as there is only one possible high-symmetry site for cation.”

We partly disagree with the reviewer about the shape of f-electron densities as a predictive tool to design highly anisotropic single-molecule magnets. As mentioned in our answers for the first round of comments, Rinehart's and Long's article is critical to the field, successfully predicting the performance of SMMs. However, it is a simplification that works well for most SMMs. The model is based on the approximation that the free electron densities of Ln³⁺ ions can be described by the quadrupole moment of f-electrons. The knowledge acquired over the last decade has shown us that the concept does not always work. For example, our group studied a series of [Er₂(COT)₃] complexes (*Chem. Commun.* 2014, 50, 1602-1604 and *J. Am. Chem. Soc.* 2014, 136, 8003-8010), in which despite the strictly axial anisotropy (that do not favour slow relaxation of magnetisation for erbium SMMs following Rinehart and Long guidelines), are still among the best performance for erbium SMMs. Other examples that prove that Rinehart and Long article do not tell the complete story are some of the pentagonal bipyramidal holmium SMMs (*Angew. Chem. Int. Ed.* 2017, 56, 11306-11308 and *Angew. Chem. Int. Ed.* 2021, 133, 27488-27493). For these systems, despite a vital charge density being present in the equatorial position (that does not favour slow relaxation of magnetisation for holmium SMMs following Rinehart and Long guidelines), SMM property is still observed.

Lines 165-166: "The increase of the magnetisation values below this temperature is likely due to a combination of thermal depopulation of the Stark sublevels" - this should say "decrease", not "increase", but also I suspect the larger decrease for the higher concentrations relative to the lower concentrations is due to stronger magnetic interactions within the NP. I note that the high temperature data, especially for the most dilute samples, decreases linearly with increasing temperature - have these data been corrected for diamagnetism?

We thank the reviewer for this comment. The reviewer is correct; it should be decreased. We have corrected the sentence. For the following sentence, "I suspect the larger decrease for the higher concentrations relative to the lower concentrations is due to stronger magnetic interactions within the NP", we completely agree with the reviewer, and indeed, this is what we analyzed in Figure 2-b. This more considerable decrease for the higher concentrations is expressed in Figure 2-b as the deviation of the linearity for the magnetization value at 1.8 K. We are also adding the data with a 1 T magnetic field which corroborates with the reviewer and also our point.

For the last question, the data were corrected for diamagnetism.

Fig 3: Why are the AC data the noisiest for sample NP10, which has the largest magnetic concentration?

We are not sure about the reason. As observed, the noise only occurs for the field-dependent ac data at higher fields. For the temperature-dependent ac data for the same sample, noise is not observed.

Figs S5-S7: The underlying Debye fits show very large alpha values, corresponding to large distributions larger than one order of magnitude in some cases. See Reta and Chilton (10.1039/C9CP04301B). Ranges corresponding to these distributions should be indicated on the plots, and discussed in the main text. I don't disagree with fitting the central tau values to obtain parameters, but it must be emphasised that these parameters are the central profiles of incredibly large distributions across the samples. Thus, statistically, I am not sure one can extract reliable information on "single" Er³⁺ sites - simply because there is not one definition of such a site, according to their magnetic data. Eg, the lines 228-230: "Reducing the dopant concentration from

4 to 1 mol% (i.e., 4 Er³⁺ ions per NP to one Er³⁺ ion per NP), a nearly 10-fold enhancement can be attained for the magnetisation relaxation times. This data further validates that SMM-like behaviour attributed to one single Er³⁺ ion can be attained at atomic level within a rigid NP matrix" are not a faithful description of the data: there is more than 100-fold variation of relaxation times within each sample, so statements like this are far too strong, and need to be appropriately tempered.

We agree with the reviewer that it is important to mention that we are using a central tau from a large alpha distribution. Following the reviewer's recommendation, we now include this information in the text. The large alpha distribution is inherent to the nature of ultrasmall NPs. With a large surface/volume ratio, decreasing the NP size can lead to a small number of doping ions close to the surface (with a distorted symmetry). This information was added to the manuscript. We also tempered our sentences substituting affirmations with suggestions.

Fig S7: I am not convinced of the need to introduce an Orbach fitting parameter for the NP1(P1) data. These P1 data should be shown on a smaller scale. However, given the large distributions in relaxation times (due to large alpha), and significant variation in CASSCF data, I do not think there is strong justification here.

We agree with the reviewer that in our previous version, the introduction of the Orbach term can be dubious. The large distribution in relaxation times with the different models in *ab initio* calculations can lead to different interpretations. However, recently, a group published EPR studies with this same system, ultrasmall cubic NaYF₄ doped with small concentrations of erbium (*Phys. Rev. B* 2022, 106, 125427). In light of this experimental work, we completely revised our calculations and benchmarked the calculated results against the published EPR study. The CASSCF calculations predict the energy barrier of 44.8 cm⁻¹ for the new [Na₈Y₄ErF₂₄]⁻ model system in which the Er³⁺ ion resides in the pseudo *O_h* symmetry (the actual point group of this model system is *D_{4h}*). The calculated value reasonably agrees with the experimentally determined $U_{eff} = 24.3 \text{ cm}^{-1}$, supporting the existence of the Orbach process. However, the calculations also show a significant QTM within the ground KD that must be quenched with the applied dc field so that the relaxation of the magnetisation could occur via other relaxation mechanisms.

Table S4-S16: Some values given in scientific notation and some not - please be consistent.

We thank the reviewer for this comment. All the values are given in scientific notation now.

Lines 235-238: "Although the data suggest that the origin of the QTM process is due to the intraparticle Er^{3+} - Er^{3+} interaction, it is not possible to completely rule out the presence of a small but non-negligible interparticle Er^{3+} - Er^{3+} interaction, as even for NP1, a small contribution in the low-frequency QTM process is still present" - examination of the fitted B1 values, which give the contribution of the QTM process, show that QTM is apparently more significant for NP1 ($16e3$) than for NP4 ($11e3$), so this argument does not hold water. Moreover, even if this statement was true, then it rather disagrees with their previous assertion that the NP1 samples contain single Er^{3+} ions. Things seem to look more sensible in the temperature-dependant data (lines 239-243, Table S17), however these data are collected at 1800 Oe where QTM should largely suppressed. Indeed, it is not clear to me that the interpretation of the data in Fig S5 are indeed QTM - this is apparently occurring over a vastly larger field range than commonly observed for QTM (usually < 1000 Oe, as seen in Fig S6).

We thank the reviewer for this comment. As we mentioned in this sentence, the presence of the low-frequency QTM process (P2) may be occurring due to inter-particle interaction, thus, explaining the presence of P2 even for NP1. Also, the observation of intra-particle QTM for NP1 does not invalidate or contradict our arguments (Process 1). We are studying nanoparticles. It is well known that NPs are not completely homogeneous in terms of composition, so we cannot exclude the presence of a small but non-negligible number of NPs containing more than 1 Er ion for NP1. This is an intrinsic issue with NPs and does not invalidate the main point of the work in showing that we can apply the knowledge acquired in the molecular field for nanomaterials.

Applying a magnetic field to reduce/eliminate QTM is a well-known strategy when aiming the QTM process intrinsic to the ion in a given crystal-field. The magnetic field results in the opening of the Zeeman sublevels, thus, removing the resonance between the level-crossings. Understanding this process allows us to rationalize that inter-molecular (herein, inter-particle) QTM cannot be circumvented *via* the application of an external magnetic field. In molecular compounds, the inter-

molecular QTM is eliminated *via* diamagnetic dilution (usually yttrium ions when studying lanthanides). This can explain the presence of P2 (much more likely to have an inter-particle nature) even at higher fields. In fact, as mentioned in the manuscript (now reworded for clarity), this was previously observed by our group for the pentagonal bipyramidal Co(II) complex (Ref. 34, *Dalton Trans.*, 2015,44, 6368-6373).

For process 1, the reviewer mentions that the B1 value is higher for NP1 (16E3) than for NP4 (11E3). We reanalyzed our data, and we noticed that we underestimated the B1 value for NP4. As can be observed in Figure 3 (left) on the main manuscript, for NP4, P1 (especially for low and high fields) occurs on the limit of our experimental frequency range, and a careful reanalysis of the data shows us an underestimation of the relaxation times in this range for NP4 (low and high fields). This resulted in a smaller-than-expected value for B1. Now, the B1 value for NP4 is 77.50E3, and the expected trend is observed.

Tables S18-24: First point: Tables S21 and S23 show all states have no symmetry, and yet all the relative angles between "main" magnetic axes are 90.00 - this is not sensible, and the data should be checked.

We thank the reviewer for raising this question. We fully agree that the reported data does not sound reasonable at first glance, but we have checked it, and it should be correct (although it is now completely removed because we overhauled and revised our calculations, see the answers below).

The point group symmetries of the model systems $[\text{Na}_6\text{Y}_8\text{ErF}_{32}]^+$ and $[\text{Na}_4\text{Y}_8\text{ErF}_{32}]^-$ are D_{2h} and C_{2h} , respectively, which are Abelian point groups. Thus, the eigenvalues (g_x , g_y , g_z) of \mathbf{g} tensor do not need to be degenerate, although the compounds have strict point group symmetries. In contrast to $[\text{Na}_6\text{Y}_8\text{ErF}_{32}]^+$ and $[\text{Na}_4\text{Y}_8\text{ErF}_{32}]^-$, the point group symmetry of $[\text{Na}_6\text{Y}_4\text{ErF}_{24}]^{3-}$ is D_{4h} which is non-Abelian point group, and it is expected that the two of its eigenvalues of \mathbf{g} tensor are identical. For example, similar results are obtained for the $[\text{ReCl}_4(\text{CN})_2]^{2-}$ single-molecule magnet which magnetic properties have been calculated in idealized D_{2h} and D_{4h} point group symmetries (<https://chemrxiv.org/engage/chemrxiv/article-details/61d57b677f1d6713383bd917>). Also, angles close to 0° (or 180°) and 90° are very typical for the systems in which Ln ion resides in a

high (pseudo)-symmetry coordination environment (for examples, see [10.1002/anie.201609685](https://doi.org/10.1002/anie.201609685) and [10.1002/chem.202003931](https://doi.org/10.1002/chem.202003931)).

In our systems, 90° angles between the main magnetic axes (Z_m) of ground and excited KDs of Er^{3+} ion also originate from the exact point group symmetries of $[\text{Na}_6\text{Y}_8\text{ErF}_{32}]^+$ and $[\text{Na}_4\text{Y}_8\text{ErF}_{32}]^-$. For example, for $[\text{Na}_4\text{Y}_8\text{ErF}_{32}]^-$, SINGLE_ANISO projects the main magnetic axis of the ground KD exactly along the z-axis of the initial cartesian coordinate system (see below). Out of three main magnetic axes (X_m , Y_m , and Z_m), Z_m is chosen by default as the quantization and main magnetic axis in SINGLE_ANISO program. Given that the main magnetic axes of the excited KDs lie in the xy-plane, expect the axis of KD2, the angle between the main magnetic axes of ground and excited KDs (3-8) will be 90° as the dot product of two vectors is zero. Below, we have given the calculated x, y, and z-components of unit vectors of the Z_m axis of each KD of Er^{3+} ion in $[\text{Na}_4\text{Y}_8\text{ErF}_{32}]^-$ which illustrate the above comment very well.

KD	x	y	z
1	0.000	0.000	1.000
2	0.000	0.000	1.000
3	-0.682	0.731	0.000
4	-0.859	-0.511	0.000
5	-0.816	0.578	0.000
6	0.997	0.071	0.000
7	-0.995	0.104	0.000
8	0.807	0.591	0.000

However, we have now completed revisited calculations (see the answer below) in light of the recently reported EPR study of cubic NaYF_4 doped with small erbium concentrations. The study showed that the ground KD of the doped NPs should be the Γ_7 doublet state ([10.1103/PhysRevB.106.125427](https://doi.org/10.1103/PhysRevB.106.125427)). In the cubic environment, this state is isotropic, and thus, we do

not report the angles between the main magnetic axes of ground KD and excited KDs for new model systems.

Overall, the calculations show a significant change in the character of the ground KD of the Er³⁺ site depending on the model. While the authors attempt to justify their discussion of the [Na₅Y₈ErF₃₂] model owing to the "convergence" of the results for the larger cluster calculations to a $mJ \pm 15/2$ ground state, they carefully omit the [Na₆Y₄ErF₂₄]₃- model from their statement, which is a large model that shows a $mJ \pm 1/2$ ground state. Further, it is the only model that approaches the true site symmetry of the compound: I am not convinced that a true high symmetry site in this compound should be able to stabilise the ca. 90% $\pm 15/2$ ground state, they claim. Clearly, symmetry AND size AND charge are important here, and things change a lot, apparently. As the magnetic data do not uniquely identify one ground state over another, I do not see justification for one choice over another. One thing the authors should check is that the quantization axes for the CFP and basis definitions is the same in all cases. A robust way to prove this, one way or the other, is using EPR spectroscopy - a low temperature CW X- or Q-band spectrum should be very diagnostic here, and greatly improve the manuscript by evidencing the magnetic ground state.

We much appreciate this criticism because it was crucial for improving the paper! We have completely revised our calculations based on the criticism and the recently reported EPR study of cubic NaY_{0.98}Er_{0.02}F₄ (*10.1103/PhysRevB.106.125427*). Thus, we again performed the *ab initio* calculations for six new model systems with varying sizes, charges, and symmetries (see ESI for further details). The model systems were chosen in such a way that their symmetry approaches the true site symmetry of the investigated NPs; the point groups of model systems belong to the subgroups of O_h . For all model systems, the same initial coordinate system (Er³⁺ ion at the origin) and quantization axis Z_m (chosen by default in the SINGLE_ANISO program) were used in calculations. The computational data obtained for the model systems still indicates that the results of the *ab initio* calculations are sensitive to the symmetry and size of the investigated model system. However, the validity of calculated results can be confirmed by benchmarking the calculated data against the previously reported data, and the charge of the model systems impacts less on the results if compared to the previous model systems. In particular, we compared the

calculated energies, g -tensors, average g values and compositions of eigenstates of each model system to the previously published data (10.1103/PhysRevB.106.125427 and 10.1016/0022-3697(62)90192-0). Before explaining the new computational results in more detail, we would also like to emphasize that we did not purposely omit any results of previous calculations; the same data was given for all model systems in ESI. The $[\text{Na}_6\text{Y}_4\text{ErF}_{24}]^{3-}$ model system with high symmetry was only used to illustrate that the investigated NPs can follow the symmetry design criteria of SMMs in the strict sense.

Out of the six newly investigated model systems, the result obtained for $[\text{Na}_8\text{Y}_4\text{ErF}_{24}]^-$ was in the best agreement with the experimental results. The *ab initio* calculations predicted almost the completely isotropic Γ_7 (KD1) ground state with the average g value of 6.79 ($g_x = 6.91$, $g_y = 6.91$, $g_z = 6.57$) for $[\text{Na}_8\text{Y}_4\text{ErF}_{24}]^-$ that is the perfect agreement with the experimental g value of 6.80 that was obtained from the EPR measurements of $\text{NaY}_{0.98}\text{Er}_{0.02}\text{F}_4$. Moreover, the order of excited states – $\Gamma_8^{(1)}$ (KD2 and KD4), Γ_6 (KD3), $\Gamma_8^{(2)}$ (KD5 and KD6), and $\Gamma_8^{(3)}$ (KD7 and KD8) – was predicted correctly for $[\text{Na}_8\text{Y}_4\text{ErF}_{24}]^-$, but it should be noted that the energy of the excited Γ_6 (KD3) doublet state is between the Kramers states (KD2 and KD4) of the first $\Gamma_8^{(1)}$ quartet state. The latter result does not contrast with the previous findings. First, García-Flores *et al.* pointed out in their paper (10.1103/PhysRevB.106.125427) that the splitting of the Kramers states of the $\Gamma_8^{(1)}$ quartet state in $\text{NaY}_{0.98}\text{Er}_{0.02}\text{F}_4$ is approximately -28 cm^{-1} . Second, Dantelle *et al.* (10.1039/B706735F) have also reported that the Γ_6 doublet state can be merged with the $\Gamma_8^{(1)}$ quartet state in the Erbium-doped $\beta\text{-PbF}_2$ single-crystals, in which the Er^{3+} ions also occupy the cubic symmetry sites. Both experimental findings support our calculated results. The only discrepancy between the previous and calculated results was that the calculations predicted the anisotropic Γ_6 (KD3) doublet state ($g_x = 7.77$, $g_y = 7.76$, $g_z = 2.19$) for $[\text{Na}_8\text{Y}_4\text{ErF}_{24}]^-$, albeit it should be isotropic in the cubic environment. However, the closer inspection of the average g value of the KD3 (5.90) and compositions of its wave functions revealed that the KD3 of the $^4\text{I}_{15/2}$ ground multiplet of Er^{3+} ion in $[\text{Na}_8\text{Y}_4\text{ErF}_{24}]^-$ can be assigned to the Γ_6 doublet state.

Taking into account all above mentioned, it can be concluded that the computational data are now in agreement with the previous (EPR) results, and we can justify which model systems best reflect the electronic structures and microscopic magnetic properties of Er^{3+} ions in the investigated NPs.

We would also like to thank the reviewer once again for this criticism which was crucial for improving the paper.

Tables S25-31: One cannot define a percentage of real and imaginary components. The contribution of a given basis function to the wavefunction is $\text{conj}(C_i) * C_i$, where C_i is the complex coefficient and $\text{conj}(C_i)$ its complex conjugate. The sum of all $\text{conj}(C_i) * C_i$ for a given state will be 1: this is the normalisation condition and thus can be represented as a percentage.

We fully agree here. We have corrected this oversight, and the caption of Tables S25-31 has been changed.

Reviewers' Comments:

Reviewer #1:

Remarks to the Author:

In this newly revised version authors, in response to both mine and ref. #4's remarks, drastically rearranged the presentation and discussion of the obtained results, and the manuscript is now much clearer in its scopes. There remain, however, a pair of important and strictly interrelated issues to be fixed before publication can be suggested.

The introduction still contains reference to "rigidity" and sparsity of the phonon spectrum as a key factor in determining slow relaxation of the magnetization. While this is in generally well-established in SMM, it is not clear how this parameter is taken into account in the course of the study. Indeed, there is no discussion on the phonon structure of the system, even by considering the model mononuclear fragment on which they carried on the ab initio studies. They added the new reference 16 on this point, which is surely relevant and timely; however, it concerns a somewhat different system and is not discussed in detail. The importance of obtaining more detailed information on the phonon spectrum of the investigated systems is even more pressing since the interpretation of the relaxation magnetic data still insists on the Orbach process via the first-excited state as a relevant point. However, as indicated both by me and reviewer #4 in our previous reports, this appears to be questionable. Indeed, despite the claim by the authors of a good agreement between calculated and experimental energy barrier, a discrepancy of about 50% between them suggests that their interpretation might not be the correct one. This is an extremely relevant point, since they propose a relaxation model through an excited state with a larger magnetization than the ground one (see fig. 4a), which is – to my knowledge – much uncommon. It is clear that, to have a firmer conclusion on the nature of the relaxation process, the calculation of the phonon spectrum of the system should be performed, following the footsteps of the results reported both in ref. 14 and in JACS2021, 143, 13633–13645). While a complete study would be outside the scope of the paper, the study of the optical phonon spectrum of the fragment alone should be feasible, for exactly the same reasons put forward by the authors to justify their study, i.e. a larger simplicity and higher symmetry of the proposed system as compared to molecular lanthanide complexes working as SMM. The inclusion of these calculations would justify the inclusion of the rigidity issue in the Introduction and in my opinion also answer to some concerns raised by referee #4 on the way phonons couple the mixed electronic states.

A few minor points to be addressed:

- a- L.170-173. There is still an issue in the description of the dc magnetic data. While the graphs report correctly MT/H, in the text authors writes that "Dc measurements are shown in Fig. 2a as magnetisation (M) versus temperature (T) from 300 K down to 1.8 K.", "an almost constant magnetisation value(...)" and "The decrease of the magnetisation values" Please change "magnetisation" in "MT/H" or equivalent expressions in all these instances.
- b- L. 174 The spin-orbit contribution is not relevant for the temperature dependence of the observed magnetic moment in lanthanide, since it is already included in the definition of the ground J multiplet: without Ligand Field splitting of the multiplet (an expression to be preferred over "Stark splitting", see the note in Rinehart-Long perspective paper for the reasons) a Curie law would be observed. Please correct.
- c- L.214-216. The described phenomenon is not new in the literature on lanthanide SMMs, and some appropriate references should be either included or recalled from those already cited.
- d- L.253-255. Authors here wrote a somehow tautological sentence: it is obvious that, on increasing the number of paramagnetic centers in a particle, both the intraparticle and the interparticle interactions will increase. This is not a suggestion from the data, as they state, but a simple consequence of the form of the interactions, which increase with the inverse of the distance. This sentence must be rearranged.
- e- In the ESI, authors should include the fit of the ac susceptibility curves as a function of frequencies, not only the derived parameters.

Reviewer #2:

Remarks to the Author:

The manuscript is greatly improved over the previous version, but there are still several points

outstanding that have not been addressed.

They still need to incorporate "ESD bars" on the magnetic relaxation plots (Figs S5-S7) to indicate the breadth of the distributions arising from the alpha values. These will be large. In the main text the authors still talk about "a nearly 10-fold enhancement" in relaxation time from NP4 - NP1, however these values will all be within one ESD of each other given the inherently large distributions. The figures need to be updated and the discussion of the data needs to correctly reflect this. Saying that large distributions move to slower timescales by a factor of 10 is OK, for example.

Once they show ESDs on the data in Fig S7, I think it will be clear that there is little justification for invoking an Orbach process here. My position is further supported by the published EPR data, showing that the ground Kramers doublet is isotropic, and thus would not be expected to show an over-barrier Orbach process. Hence, they should remove the Ueff fitting part and discussion. There is nothing wrong with QTM/Direct/Raman processes though, and moreover, there is nothing wrong comparing the electronic states from their calculations to the energy spectra derived experimentally.

In their reply letter, they state "Also, the observation of intra-particle QTM for NP1 does not invalidate or contradict our arguments (Process 1). We are studying nanoparticles. It is well known that NPs are not completely homogeneous in terms of composition, so we cannot exclude the presence of a small but non-negligible number of NPs containing more than 1 Er ion for NP1. This is an intrinsic issue with NPs and does not invalidate the main point of the work in showing that we can apply the knowledge acquired in the molecular field for nanomaterials." - The magnetic data they observe for QTM in NP1 in process 1 cannot be simply described as arising from "a small but non-negligible number of NPs containing more than 1 Er ion", unless all of the magnetic data for NP1 is also described as such, which I do not think they are trying to do here. So I do not buy their argument. While in the revised paper they have more fully mentioned the intra-particle QTM for NP1 (lines 234-239), they have left some lines in that are not in agreement with their new words. Lines 269/270 "However, for NP1, removing the intraparticle dipolar interaction results in a thermally activated relaxation barrier..." - intraparticle dipolar interactions have not been removed in NP1. Line 327/328 "we successfully demonstrated that slow magnetic relaxation could be attained for a single Er³⁺ ion within a nanoparticle matrix." - a single Er³⁺ ion has not been isolated.

I must say that the computational work is now much stronger, and given the benchmarking to EPR, it is much more robustly presented.

My last request is that they introduce the existing experimental EPR work on the same Er@NaYF₄ NPs (now crucial to their work) in the introduction (somewhere between lines 74 and 117).

Small comments:

- line 132/133 "any distortions on the Oh site can increase the magnetic anisotropy" -> "any distortions at the Oh site can **give rise to** magnetic anisotropy"

- line 191 "the dipolar coupling at $\sim 8 \text{ \AA}$ is two orders of magnitude smaller than at $\sim 3 \text{ \AA}$." -> "the dipolar coupling at $\sim 8 \text{ \AA}$ is **one order** of magnitude smaller than at $\sim 3 \text{ \AA}$."

Answers to Reviewer's Comments

REVIEWER COMMENTS:

Reviewer #1 (Remarks to the Author):

In this newly revised version authors, in response to both mine and ref. #4's remarks, drastically rearranged the presentation and discussion of the obtained results, and the manuscript is now much clearer in its scopes. There remain, however, a pair of important and strictly interrelated issues to be fixed before publication can be suggested.

We thank the reviewer for these helpful and relevant comments and based on them, new fitting of the magnetic data, and periodic boundary calculations carried out for α -NaYF₄ at the DFT level; we revisited the discussion related to the scope of the paper and magnetic relaxation mechanisms of Er-doped NPs.

The introduction still contains reference to "rigidity" and sparsity of the phonon spectrum as a key factor in determining slow relaxation of the magnetization. While this is in generally well-established in SMM, it is not clear how this parameter is taken into account in the course of the study. Indeed, there is no discussion on the phonon structure of the system, even by considering the model mononuclear fragment on which they carried on the ab initio studies. They added the new reference 16 on this point, which is surely relevant and timely; however, it concerns a somewhat different system and is not discussed in detail.

To testify to the "rigidity" of α -NaYF₄, we performed the periodic boundary calculations for α -NaYF₄ at the DFT level using the CRYSTAL17 program (see ESI for details). In particular, we calculated the phonon dispersion relations, density of states (DOS), and projected DOS for α -NaYF₄ and α -NaY^{167.259}F₄. The latter model system, in which the mass of Y (88.906) was replaced with the mass of Er³⁺ ion (167.259), simulates the effect of Er³⁺ ion on the phonon spectrum of α -NaYF₄. Because the CRYSTAL17 cannot model 50/50 occupancy of Na⁺ and Y³⁺ ions on the octahedral site of the cubic lattice (Fm3m, *O_h*), all calculations were performed for the model systems consisting of alternating layers of Na⁺ and Y³⁺ ions in the tetragonal (P4/mmm, *D_{4h}*) point group. In other words, the positions of Na⁺ and Y³⁺ ions in the conventional cubic cell (1 × 1 × 1) cannot be chosen in such a way that the cubic symmetry is retained. During the study, we noticed that reducing the symmetry of model systems to the tetragonal one was mandatory because any attempt to optimize the atomic positions and unit cells of 2 × 2 × 2 supercells of α -NaYF₄ retaining

the cubic symmetry led to the solutions that were not minimum on the potential energy hypersurface or not converged in the set convergence criteria.

As explained in the ESI, the calculated phonon dispersion relations and DOS for α -NaYF₄ show that the first optical phonon mode of α -NaYF₄ is observed at 125 cm⁻¹ below which the phonon DOS is dictated by acoustic phonons, whereas in the region of 125–177 cm⁻¹ both, acoustic and optical, phonons contribute to the phonon DOS, and above 177 cm⁻¹ optical phonons dominate the phonon DOS (Figure S12). It also can be seen from Figure S12 that α -NaYF₄ has two different regions, ~170–260 cm⁻¹ and ~295–360 cm⁻¹, where the accumulation of phonons occurs. Otherwise the phonon DOS is low compared to high density regions, and all high energy lattice vibrations that are typically observed for molecules are absent. Similar analysis can also be performed for α -NaY^{167.259}F₄ (Figure S13). The obtained results can be compared to molecular compounds in which lattice vibrations expand from 0 cm⁻¹ to 3500 cm⁻¹ and phonon DOS shows several regions where the density of phonons is high (10.1021/jacs.1c05068, 10.1002/adom.202201675, 10.1002/adom.202101721). Whilst the above results do not indicate how strongly or weakly each lattice vibrations of α -NaYF₄ are coupled to the spin of Er³⁺ ions, the results indicate that NPs based on α -NaYF₄ have fewer vibrational modes, which could couple with spins, and sparser phonon spectrum than molecular compounds. Thus, the NPs based on α -NaYF₄ can function as alternatively host materials for paramagnetic ions exhibiting slow relaxation of the magnetisation. Furthermore, it is important to note that their vibration bands (and phonon spectrum) can be tuned in a rational manner by changing the nature and concentration of doped atoms from one to other, as exemplified by the previously reported data for NaYF₄ (For example, see 10.1039/D3RA03148A, 10.1039/C3CE40362A, 10.1038/srep29871, 10.1021/acsami.7b05480, 10.1007/s12034-023-02978-4, 10.1016/j.jallcom.2008.07.050, 10.1021/cm2004227, 10.1016/j.jlumin.2005.03.011, 10.1016/j.jallcom.2020.157631, 10.1039/D3RA02841K, 10.1016/j.apsusc.2013.02.026).

The importance of obtaining more detailed information on the phonon spectrum of the investigated systems is even more pressing since the interpretation of the relaxation magnetic data still insists on the Orbach process via the first-excited state as a relevant point. However, as indicated both by me and reviewer #4 in our previous reports, this appears to be questionable. Indeed, despite the claim by the authors of a good agreement between calculated and experimental energy barrier, a discrepancy of about 50% between them suggests that their interpretation might not be the correct one. This is an extremely relevant point, since they propose a relaxation model through an excited state with a larger magnetization than the ground one (see fig. 4a), which is – to my knowledge – much uncommon.

In terms of the percent error, we can speak here of a large error; however, given that we are speaking about the energies that are very small, the calculated and experimental ones correlate reasonably well. For example, if we would compare the experimental and calculated energies of higher lying KD whose energy, for example, is 300 cm^{-1} and the difference between the experimental and calculated value is 20 cm^{-1} , the percent error would be significantly smaller, only $\sim 6\%$. Nevertheless, we have fully revised the magnetic data, and based on the new fits, we propose that the relaxation of the magnetization proceed through the QTM and Raman processes and the Orbach process does not play a role in relaxation. This analysis is further supported by the larger magnetization of the first excited KD compared to the ground one, as stated by the referee, and the fact that the ground KD of Er^{3+} ion is isotropic.

It is clear that, to have a firmer conclusion on the nature of the relaxation process, the calculation of the phonon spectrum of the system should be performed, following the footsteps of the results reported both in ref. 14 and in JACS2021, 143, 13633–13645). While a complete study would be outside the scope of the paper, the study of the optical phonon spectrum of the fragment alone should be feasible, for exactly the same reasons put forward by the authors to justify their study, i.e. a larger simplicity and higher symmetry of the proposed system as compared to molecular lanthanide complexes working as SMM. The inclusion of these calculations would justify the inclusion of the rigidity issue in the Introduction and in my opinion also answer to some concerns raised by referee #4 on the way phonons couple the mixed electronic states.

We agree with the review that the detailed spin-phonon coupling calculations would have shed light on the magnetic relaxation mechanisms of the studied Er-doped NPs. Still given the new information we have from the studied Er-doped NPs and computational costs of spin-phonon coupling calculations, in our opinion, the detailed *ab initio* spin-phonon coupling calculations for $[\text{Na}_8\text{Y}_4\text{ErF}_{24}]^-$ are not necessary and are out of scope this study for the three reasons.

First, in light of the new fitting of the magnetic data, it is highly unlikely that the Orbach process contributes to the slow relaxation of magnetisation in the Er-doped NPs, as mentioned above, and the relaxation mechanism is now explained much better with the existing experimental magnetic data and calculations.

Second, when done from the first principles using *ab initio* methods, the spin-phonon calculations, require a large amount of computational time and resources. Lunghi, Chilton *and co-workers* have done ground-breaking research by introducing a new approach to treat the spin-phonon coupling either by using the phenomenological parameters (10.1103/PhysRevB.101.174402) or by treating everything from the first principles (for example

see. 10.1021/jacs.1c05068 and 10.1021/jacs.1c01410). Layfield & Mansikkamäki *and co-workers*. have also reported an *ab initio* approach from the first principles that takes into account the spin-phonon couplings arising from optical phonons (10.1126/science.aav0652). We have used this method to calculate the spin-phonon couplings of other molecular system in another ongoing study. If this method, like any other *ab initio* methods, is applied to calculate the spin-phonon couplings arising from the optical phonons for the fragment $[\text{Na}_8\text{Y}_4\text{ErF}_{24}]^-$, we would need to run an extensive amount of the CASSCF calculations. The fragment $[\text{Na}_8\text{Y}_4\text{ErF}_{24}]^-$ has 105 normal vibration modes, and just by taking into account the first-order contributions of optical phonons (Orbach and direct processes), we would need to calculate the first derivatives of the crystal field parameters with respect to each of normal modes of the fragment $[\text{Na}_8\text{Y}_4\text{ErF}_{24}]^-$ (the full Hamiltonian is given in ESI of 10.1126/science.aav0652). To have enough accurate results, we have found that five displacements are enough along the positive and negative direction of each normal mode and these displacements must be generated along each axis (x, y, z). This would sum up 3150 individual CASSCF calculations in the case of the fragment $[\text{Na}_8\text{Y}_4\text{ErF}_{24}]^-$! Furthermore, because slow relaxation of the magnetization in the studies Er-doped NPs most likely proceeds *via* Raman and QTM mechanisms in light of the new results, we should also include the second-order contributions of optical phonons (Raman process) into the spin-phonon calculations that are obtained from the second derivatives of the crystal field parameters. Thus, the amount of needed CASSCF calculations, even for the optical phonons, increases very quickly. This is the main reason why the *ab initio* spin-phonon calculations are not routinely performed even for the small (e.g. mononuclear) molecular systems.

Third, the fragment $[\text{Na}_8\text{Y}_4\text{ErF}_{24}]^-$ is a problematic case for the spin-phonon coupling calculations. It is directly extracted from the crystal structure of $\alpha\text{-NaYF}_4$ and quick geometry optimization at the DFT level shows that the geometry of $[\text{Na}_8\text{Y}_4\text{ErF}_{24}]^-$ is not retained in the gas phase. This means it is not a minimum energy structure on the potential energy hypersurface, which is the first requirement for the spin-phonon coupling calculations. The vibrational spectrum of an investigated compound cannot contain any imaginary frequencies if the spin-phonon calculations are aimed for. Despite this, we wish to point out that the $[\text{Na}_8\text{Y}_4\text{ErF}_{24}]^-$ fragment can be used in the CASSCF calculations to evaluate the microscopic magnetic properties of the Er-doped NPs. The extraction of the model structure from the solid-state structure to the CASSCF calculations is a well-established procedure in the field and usually yields very good results, as we have shown here by comparing the calculated values to the experimental ones obtained from the EPR data of $\text{NaY}_{0.98}\text{Er}_{0.02}\text{F}_4$. In principle, we could construct new fragment from the periodic boundary calculations to the spin-phonon coupling calculations and model the charges of surrounding atoms by the Madelung potential, but as mentioned above, in our opinion, the detailed *ab initio* spin-phonon coupling calculations are out of scope of this study.

Considering all the above mentioned, we hope that the referee understands our decision not to conduct the spin-phonon coupling calculations for the fragment $[\text{Na}_8\text{Y}_4\text{ErF}_{24}]^-$. Instead, we performed the detailed periodic boundary calculations at the DFT level for $\alpha\text{-NaYF}_4$ and $\alpha\text{-NaY}^{167.259}\text{F}_4$ to understand the phonon spectra of the investigated Er doped NPs that shed light on the “rigidity” of the studies Er-doped NPs as well as providing an approach to evaluate at the qualitative level which of the calculated vibrational modes of Er@NaYF_4 might couple to the spins of Er^{3+} ions (see ESI). The latter can be performed as the lattice vibrations that are associated with magnetic centers usually trigger the largest spin-phonon couplings, although this is not always the case (10.1021/jacs.1c05068). We have explicitly stated the limitations of this qualitative analysis in ESI and mentioned that the results of this analysis are only approximate.

A few minor points to be addressed:

a- L.170-173. There is still an issue in the description of the dc magnetic data. While the graphs report correctly MT/H, in the text authors writes that “Dc measurements are shown in Fig. 2a as magnetisation (M) versus temperature (T) from 300 K down to 1.8 K.”, “an almost constant magnetisation value(…)” and “The decrease of the magnetisation values” Please change “magnetisation” in “MT/H” or equivalent expressions in all these instances.

Answer: We thank the reviewer for this point. We are now discussing in terms of mass susceptibility ($\chi_g = \text{emu g}^{-1}$).

b- L. 174 The spin-orbit contribution is not relevant for the temperature dependence of the observed magnetic moment in lanthanide, since it is already included in the definition of the ground J multiplet: without Ligand Field splitting of the multiplet (an expression to be preferred over “Stark splitting”, see the note in Rinehart-Long perspective paper for the reasons) a Curie law would be observed. Please correct.

Answer: We corrected the sentence accordingly to the reviewer suggestion.

c- L.214-216. The described phenomenon is not new in the literature on lanthanide SMMs, and some appropriate references should be either included or recalled from those already cited.

Answer: We thank the reviewer for this comment. Appropriate references are now added (37-41).

d- L.253-255. Authors here wrote a somehow tautological sentence: it is obvious that, on

increasing the number of paramagnetic centers in a particle, both the intraparticle and the interparticle interactions will increase. This is not a suggestion from the data, as they state, but a simple consequence of the form of the interactions, which increase with the inverse of the distance. This sentence must be rearranged.

Answer: As requested, we rearranged the sentence to *“This is in agreement with the expectations since not only intraparticle interactions are affected when more Er^{3+} ions are confined within a single NP, but also the probability of interparticle interaction increases.”*

e- In the ESI, authors should include the fit of the ac susceptibility curves as a function of frequencies, not only the derived parameters.

Answer: The ac susceptibility curves as a function of frequencies are shown in the main manuscript (Figure 3). We don't think it is necessary to repeat it on the ESI.

Reviewer #2 (Remarks to the Author):

The manuscript is greatly improved over the previous version, but there are still several points outstanding that have not been addressed.

They still need to incorporate "ESD bars" on the magnetic relaxation plots (Figs S5-S7) to indicate the breadth of the distributions arising from the alpha values. These will be large. In the main text the authors still talk about "a nearly 10-fold enhancement" in relaxation time from NP4 - NP1, however these values will all be within one ESD of each other given the inherently large distributions. The figures need to be updated and the discussion of the data needs to correctly reflect this. Saying that large distributions move to slower timescales by a factor of 10 is OK, for example.

Answer: We thank the reviewer for all the suggestions to improve the quality of our work. We have included the ESD bars for all the plots (Figs. S5-S7). We modified this sentence to describe our observations better.

Once they show ESDs on the data in Fig S7, I think it will be clear that there is little justification for invoking an Orbach process here. My position is further supported by the published EPR data, showing that the ground Kramers doublet is isotropic, and thus would not be expected to show an over-barrier Orbach process. Hence, they should remove the Ueff fitting part and discussion.

There is nothing wrong with QTM/Direct/Raman processes though, and moreover, there is nothing wrong comparing the electronic states from their calculations to the energy spectra derived experimentally.

Answer: Following the reviewer's suggestion and the analysis of all the data with the ESD bars, we removed the Orbach process from our analysis.

In their reply letter, they state "Also, the observation of intra-particle QTM for NP1 does not invalidate or contradict our arguments (Process 1). We are studying nanoparticles. It is well known that NPs are not completely homogeneous in terms of composition, so we cannot exclude the presence of a small but non-negligible number of NPs containing more than 1 Er ion for NP1. This is an intrinsic issue with NPs and does not invalidate the main point of the work in showing that we can apply the knowledge acquired in the molecular field for nanomaterials." - The magnetic data they observe for QTM in NP1 in process 1 cannot be simply described as arising from "a small but non-negligible number of NPs containing more than 1 Er ion", unless all of the magnetic data for NP1 is also described as such, which I do not think they are trying to do here. So I do not buy their argument. While in the revised paper they have more fully mentioned the intra-particle QTM for NP1 (lines 234-239), they have left some lines in that are not in agreement with their new words. Lines 269/270 "However, for NP1, removing the intraparticle dipolar interaction results in a thermally activated relaxation barrier..." - intraparticle dipolar interactions have not been removed in NP1. Line 327/328 "we successfully demonstrated that slow magnetic relaxation could be attained for a single Er³⁺ ion within a nanoparticle matrix." - a single Er³⁺ ion has not been isolated.

Answer: We thank the reviewer for insisting on this specific point, which allowed us to improve the quality of the manuscript. Given the complexity of the system, and due to the pioneering study with NPs, i.e., there is no literature to compare with other NP systems, we opted to tone down and discuss the difficulties in assigning the nature of process 1 QTM. We now discuss that even with 1 Er³⁺ particle, the distance is reasonable for Er³⁺-Er³⁺ interactions in some specific conditions. We are also keeping the discussion about the possible non-homogeneity intrinsic from NP systems. Currently, the text is written as follows:

"For process 1 QTM, it is not possible to unequivocally determine if the nature of the process is due to intraparticle, interparticle, or a combination of both. For example, in NP4, the mean distance between the Er³⁺ ions was determined as 9.1 Å, while for NP2, the mean distance between the two Er³⁺ ions was determined as 12.9 Å. If we assume two NP1 aggregated side-by-side (see TEM image, Figure 1a), containing only one Er³⁺ ion, and with the Er³⁺ ion perfectly

positioned in the center of the NP, the distance between these Er^{3+} ions will be 26.9 Å. If the ions at two neighboring NPs are not located in the centre, but facing each other, the distance between the Er^{3+} ions can be even smaller. At these distances, the presence of the process 1 QTM can be expected, and the lower B1 value, when compared to NP2 and NP4, is in agreement with the increased distancing. Additionally, the observation of intraparticle-mediated QTM for NP1 cannot be fully ruled out. Although our data and calculations suggest the presence of only one Er^{3+} per NP for NP1, NPs are not perfectly homogeneous in terms of composition. Thus, one can expect that a small but non-negligible amount of NPs contains more than one Er^{3+} ion within the structure. Hence, unequivocally assignment of the process 1 QTM is not possible since both intra- and interparticle interactions can be reasonably explained based on the NP structural features."

I must say that the computational work is now much stronger, and given the benchmarking to EPR, it is much more robustly presented.

My last request is that they introduce the existing experimental EPR work on the same $Er@NaYF_4$ NPs (now crucial to their work) in the introduction (somewhere between lines 74 and 117).

Answer: We thank the reviewer for the comment and suggestion. We understand the reason for the referee request; however, in our opinion introducing the experimental EPR work close to our theoretical discussion make it easier for the readers. As the EPR is not our work, neither our goal with this study, we do not see a real necessity to discuss it on the introduction.

Small comments:

- line 132/133 "any distortions on the Oh site can increase the magnetic anisotropy" -> "any distortions at the Oh site can **give rise to** magnetic anisotropy".
- line 191 "the dipolar coupling at ~ 8 Å is two orders of magnitude smaller than at ~ 3 Å." -> "the dipolar coupling at ~ 8 Å is **one order** of magnitude smaller than at ~ 3 Å."

Answer: We thank the reviewer for the suggestion. We reworded the sentence accordingly to the reviewer suggestion.

Reviewers' Comments:

Reviewer #1:

Remarks to the Author:

In the third revision of this work authors satisfactorily solved the two most critical points which remained to be tackled, namely the inclusion of theoretical calculation of the phonon spectrum of the doped nanoparticles and a complete reconsideration of their magnetization relaxation processes, as suggested by both me and reviewer #2. Thus, even if some of the results and interpretation provided in the paper may remain open to discussion, I consider the manuscript to be now acceptable for publication in Nature Communications, given its broad interest and the appropriate methodology used.

Reviewer #2:

Remarks to the Author:

The authors have improved the manuscript following advice from both referees. I am satisfied that they have addressed my comments. I am also particularly impressed with the thorough phonon calculations, which look very well done and are well-explained in the SI. These greatly enhance the quality of the work.